# Mesoscale DNA features impact APOBEC3A and APOBEC3B deaminase activity and shape tumor mutational landscapes

Ambrocio Sanchez [1,2], Pedro Ortega [1,2], Ramin Sakhtemani [3,4], Lavanya Manjunath [1,2], Sunwoo Oh [1,2], Elodie Bournique [1,2], Alexandrea Becker[1,2], Kyumin Kim [5], Cameron Durfee [6], Nuri Alpay Temiz [7,8], Xiaojiang S. Chen [5], Reuben S. Harris [6,9], Michael S. Lawrence[3,4,10] & Rémi Buisson [1,2,11] ✉

Antiviral DNA cytosine deaminases APOBEC3A and APOBEC3B are major sources of mutations in cancer by catalyzing cytosine-to-uracil deamination. APOBEC3A preferentially targets single-stranded DNAs, with a noted affinity for DNA regions that adopt stem-loop secondary structures. However, the detailed substrate preferences of APOBEC3A and APOBEC3B have not been fully established, and the specific influence of the DNA sequence on APO-BEC3A and APOBEC3B deaminase activity remains to be investigated. Here, we find that APOBEC3B also selectively targets DNA stem-loop structures, and they are distinct from those subjected to deamination by APOBEC3A. We develop Oligo-seq, an in vitro sequencing-based method to identify specific sequence contexts promoting APOBEC3A and APOBEC3B activity. Through this approach, we demonstrate that APOBEC3A and APOBEC3B deaminase activity is strongly regulated by specific sequences surrounding the targeted cytosine. Moreover, we identify the structural features of APOBEC3B and APOBEC3A responsible for their substrate preferences. Importantly, we determine that APOBEC3B-induced mutations in hairpin-forming sequences within tumor genomes differ from the DNA stem-loop sequences mutated by APOBEC3A. Together, our study provides evidence that APOBEC3A and APOBEC3B can generate distinct mutation landscapes in cancer genomes, driven by their unique substrate selectivity.

The Apolipoprotein B mRNA-editing enzyme catalytic polypeptide-like (APOBEC) proteins promote the deamination of cytosine to uracil in DNA or RNA[1–3]. APOBEC enzymes serve as essential components of the immune system, acting as defense mechanisms against DNA or RNA viruses and transposons by inducing mutations[2,4,5]. However, APOBEC proteins are also one of the predominant causes of genomic mutations in cancer, and recent cancer-focused genomic studies have identified APOBEC-associated mutations in >70% of cancer types[6–17]. These mutations are particularly prevalent in breast, lung, cervical, and head & neck cancer genomes[11,15,18,19]. APOBEC mutations display a non-uniform distribution across cancer genomes, showing a bias towards the lagging strand template of the DNA replication fork and forming hypermutation clusters known as kataegis and omikli[20–27]. Two of the eleven members of the APOBEC family, APOBEC3A (A3A) and APO-BEC3B (A3B) are responsible for the majority of the APOBEC muta-tional signatures identified in tumor cells[6,8,10,15–17,19,27,28]. Both A3A and

A3B are present in the nucleus causing mutations in cell genomes in addition to their normal function of protecting cells against viral infections[8,9,29,30]. The ability of A3A and A3B to rewrite genomic information has established them as significant drivers of diversity and heterogeneity within tumor genomes[1,31–33]. In addition to causing mutations, overexpression of both A3A and A3B in cancer cells results in an increase in replication stress and the formation of DNA double-strand breaks[8,29,31,34–36]. Finally, emerging evidence has implicated A3A and A3B in the promotion of cancer drug resistance, further underscoring their impact on disease progression[28,37].

A3B is highly expressed across a wide range of tumor types, including breast, lung, colorectal, bladder, cervical, head & neck, and ovarian cancer[1,7,8,10,35,38]. A3A is also expressed in those tumor types, but in fewer patients and at lower levels[8,15,34]. A3A expression in cancer cells is transiently regulated, triggered by various cellular stresses encountered by the cells, leading to episodic bursts of mutations[6,19,28,39], explaining the poor correlation between A3A expression levels and A3A-associated mutations in individual tumor samples. Moreover, the enzymatic activity of A3B is weaker than A3A which allows the cells to tolerate higher levels of A3B[40–42]. This disparity in enzymatic activity possibly explains the higher prevalence of A3B expression found in tumors, whereas prolonged expression of A3A is detrimental for cancer cells[1,8,29,43].

A3A and A3B both target TpC motifs on single-stranded DNA (ssDNA) to promote the deamination of cytosine to uracil (C > U). However, A3A and A3B exhibit distinct substrate preferences. A3A favors cytidine deamination on a YTC motif, whereas A3B prefers an RTC sequence motif (where Y is a pyrimidine and R is a purine)[44]. Moreover, A3A and A3B can deaminate RNA substrates[18,42,45–47]. Recent research conducted in our laboratory revealed that A3A targets specific DNA stem-loop structures in the genomes of tumor cells[15]. These DNA stem-loops mutated by A3A are not random and display distinct patterns. A3A preferentially deaminates TpC that are present in hairpin structures featuring 3- or 4-nucleotide (nt) loops with a cytosine located at the 3′ position of the loop[15,16,48–50]. In addition, the sequence of the loop itself significantly influences A3A's ability to catalyze deamination[15,16]. These substrate preferences can be utilized to identify tumors with A3A-driven mutations[15,16,18,28]. However, whether A3B also targets DNA substrates with specific structural characteristics remains unclear.

A3A is a 23 kDa protein formed of a single domain, whereas A3B (46 kDa) is composed of two structurally homologous domains. A3B's catalytic activity resides in its C-terminal domain (CTD) which shares 92% amino acid identity with A3A[51]. A3B N-terminal domain (NTD) is known to bind DNA and RNA[52], promote A3B nuclear localization[30,53,54], and facilitate A3B enzymatic activity and processivity[52,55]. The difference in enzymatic activity relies on specific structural differences around the active sites of A3A and A3B. The interaction between ssDNA and A3A or A3B^CTD is mediated by the loops 1, 3, and 7. Multiple studies have shown that the substitution of A3B loop 1 (DPLVLRRRQ) with the A3A loop 1 sequence (GIGRHK) results in a strong increase in the deaminase activity due to a significant structural change that causes A3B's active site to transition from a closed to an open conformation[17,40,41,56–58]. In contrast, loop 3 of A3A and A3B varies by merely one amino acid, while loop 7 is identical. In fact, loop 7 plays a crucial role in determining the preference for cytosine bases preceded by a thymine[40,57]. Nevertheless, it is still unclear how the interaction between ssDNA and the loops impacts the preference of A3A and A3B for certain types of DNA secondary structures.

In addition to their role in promoting cancer mutagenesis and antiviral functions, both A3A and A3B were used to produce base editing tools[59–62]. Base editing technologies have revolutionized the potential to correct genetic diseases by generating specific and precise point mutations in genomic DNA[63,64]. Base editing consists of the recruitment of a DNA cytosine (APOBEC1, A3A, A3B, or AID) or an adenosine (TadA) deaminase to a defined location in the genome by using components of the CRISPR systems. DNA binding of the Cas9-guide RNA ribonucleoprotein complex forms an R-loop structure with ssDNA that is exposed to the deamination activity of the enzymes fused to the Cas9[65,66]. The deamination rate of the target nucleotide is strongly impacted by surrounding DNA secondary structure features limiting the efficiency of the base editors[65]. Therefore, it is essential for the development of more efficient base editing tools to better understand how A3A, A3B, and other deaminase enzymes target specific DNA secondary structures and DNA sequences.

In this study, we find that similar to A3A, A3B preferentially targets DNA stem-loop structures. We develop a sequencing-based in vitro assay to identify sequence contexts preferentially targeted by A3A and A3B. We show that A3B exhibits a preference for deaminating DNA hairpins that possess 4- or 5-nucleotide loops with specific sequences surrounding the TpC motif. These findings contrast with the preferred substrates of A3A, which predominantly target DNA stem-loop structures with smaller loops of three nucleotides. Moreover, we identify the specific amino acids on A3A and A3B that are responsible for their substrate selectivity. Importantly, we find evidence of A3B-induced DNA stem-loop mutations in mouse and human tumor genomes. Collectively, our data suggest that the differential activities of A3B and A3A will result in distinct mutational landscapes within cancer cells.

## Results

### APOBEC3B targets specific DNA stem-loop structures

Structural studies of A3A, A3B, and A3G in complex with ssDNA revealed a U-shaped conformation of the DNA when bound to the active site[40,57,67] (Fig. 1a, b and Supplementary Fig. 1A). The superposition of all three structures emphasized that the ssDNA in complex with A3A and A3B forms a tight U-turn with the surrounding nucleotides that can make base pairing contacts due to their close proximity. In contrast, the ssDNA bound to A3G revealed a distinct orientation with the nucleotides before and after the cytosine pointing in the opposite direction, thereby hindering the formation of potential DNA stem-loop structures (Supplementary Fig. 1B). It is important to point out that several mutations were made in A3B to help solubilize the protein and stabilize its association with ssDNA[40]. The mutations on A3B were derived from A3A by switching the amino acid sequence of A3B loop 1 with A3A loop 1. In addition, loop 3 of A3B was partially removed (Fig. 1b). Therefore, it is still unclear how ssDNA binds to wild-type A3B. Structural prediction of full-length A3B highlights how the three loops surrounding the active site create a deep pocket that can be only accessed by ssDNA with a U-turn conformation (Fig. 1c)[41,56]. This suggests that U-shaped structures already formed in DNA stem-loops are likely more favorable than linear DNA for deamination by A3B.

To investigate substrate preferences of A3B, we used a cell-free in vitro biochemical assay to measure the efficiency of cytosine deamination by A3A and A3B on synthetic DNA substrates derived from a recurrent mutated TpC site in the NUP93 gene. This hairpin-forming sequence was previously identified to be mutated in several patient tumor samples with high levels of APOBEC mutations[15]. When the cytosine in the TpC motif is deaminated, the resulting U is removed by the action of purified uracil DNA glycosylase (UDG) in the reaction buffer. The abasic site (AP site) undergoes site-specific breakage under alkaline conditions at 95 °C, and the cleavage product can be visualized and quantified with near-nucleotide resolution by electrophoresis under denaturing conditions (Fig. 1d). To specifically monitor A3B deaminase activity, we selected U2OS cells that express high levels of endogenous A3B but not A3A[18]. Conversely, A3A was expressed in HEK-293T cells that lack A3A and A3B expression (Supplementary Fig. 1C, D). We and others previously demonstrated that these cell-free systems recapitulate activity observed with recombinant proteins[8,15,18,49,52,55,68–70], but also with the advantage of measuring

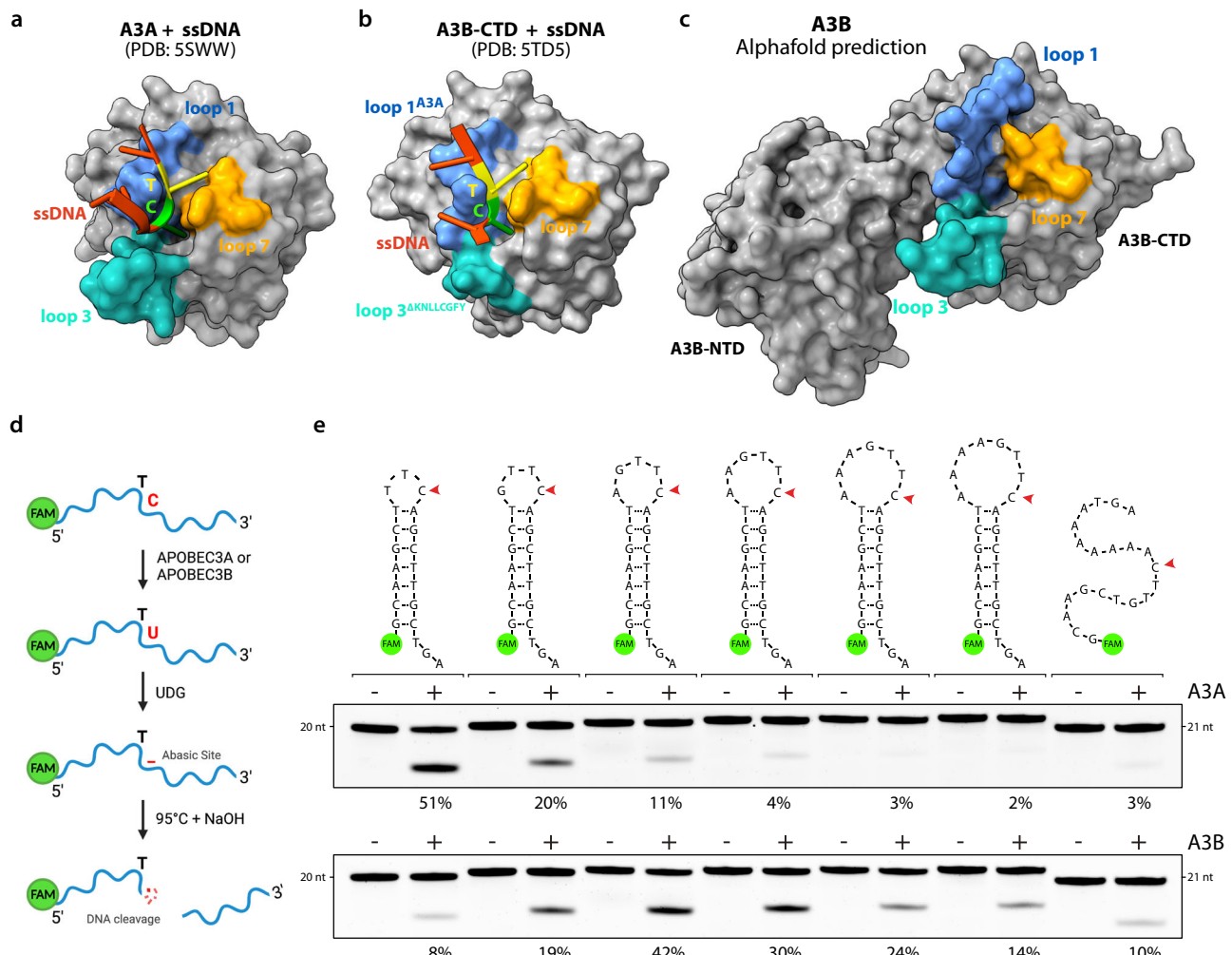

**Fig. 1 | APOBEC3A and APOBEC3B preferentially target U-shaped DNA.**
**a**, **b** Molecular surface of the A3A and A3B-CTD in complex with ssDNA. The loops surrounding the active site have been color-coded as shown. PDB IDs shown for reference. **c** Molecular surface of A3B full-length predicted with AlphaFold Protein Structure Database. **d** Schematic of the in vitro assay for APOBEC cytidine deamination activity. Created with BioRender.com. **e** Deamination activity was monitored on the indicated DNA stem-loop substrates using 1 µg of HEK-293T whole cell extract expressing A3A or 30 µg of U2OS whole cell extract expressing endogenous A3B. The percentage of cleavage is indicated. Source data are provided as a Source Data file.

enzymatic activity in a more physiologically relevant context. We first monitored A3A and A3B deaminase activity on a set of oligonucleotides that form hairpin structures with increasing loop sizes from 3 nt to 8 nt or linear ssDNA. Note that a minimum of 3-nt loop was required for the folding of the hairpins[71]. As we previously demonstrated, A3A has a strong preference for hairpins with a 3-nt loop, and deamination activity mediated by A3A decreased with the expansion of the loop size (Fig. 1e)[15]. Surprisingly, we found that A3B also targets DNA stem-loop structures. However, A3B showed preference for hairpins with intermediate-sized loops from 4 to 6 nt and disliked shorter or longer loops. Importantly, A3B displayed a clear preference for a 5 nt hairpin loop over linear ssDNA (Fig. 1e). We then validated that the measured deamination activities were exclusively dependent on A3A or A3B respectively (Supplementary Fig. 1E, F). Moreover, we verified the formation of the stem-loop structures by exonuclease T (ExoT) assay[15], which cleaves ssDNA from its 3' end. We showed that unfolded ssDNA was completely degraded in the presence of Exo T, whereas only the 3' ssDNA tails of the stem-loop structures were cleaved by Exo T (Supplementary Fig. 1G). Finally, to demonstrate that our assay was not rate-limited by UDG activity that might be affected by the structures of these substrates, we tested synthetic uracil-containing DNA substrates, both ssDNA and hairpin, and all were fully cleaved under the assay

conditions (Supplementary Fig. 1H), establishing that this assay provides a faithful readout of A3A and A3B activity. Together, these results demonstrate that A3B, similar to A3A, exhibits an ability to target DNA stem-loop structures. However, A3B displays a distinct preference for stem-loop structures with longer loops compared to A3A.

## Oligo-seq, a sequencing-based method to define optimal substrates of APOBEC3B and APOBEC3A

To interrogate how A3A and A3B activities are impacted by mesoscale genomic features—characterized by DNA sequences ranging from 3- to 30-base pair length with the capacity to adopt various structural configurations, we aimed to use an unbiased approach by developing a sequencing-based method we named Oligo-seq to identify the sequences deaminated by A3A or A3B (Fig. 2a). We first designed a small 20-nt oligonucleotide that forms a hairpin, contains a single TpC motif in 3' position of a 3-nt loop, and a random nucleotide in position −2 (relative to C in position 0) that was flanked by a 5-basepair (bp) stem (Fig. 2b). We opted for a 5-bp stem hairpin to limit any potential inhibition of the DNA polymerase used in the subsequent step of the Oligo-seq method. Common sequencing library methods are not adapted for the sequencing of such short DNA oligonucleotides. Typically, these methods require the addition of an adapter by ligation

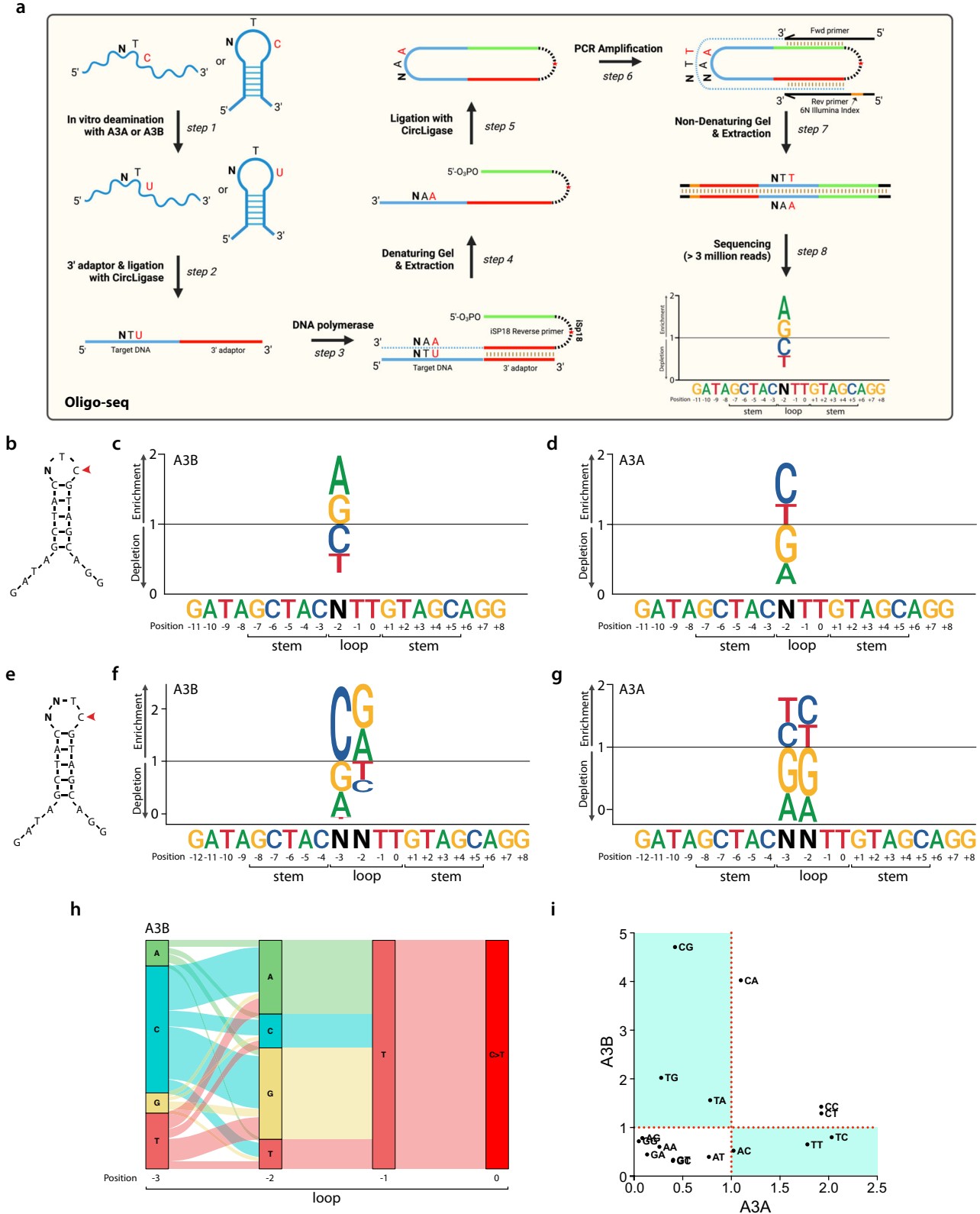

**Fig. 2 | Oligo-seq, a sequencing-based in vitro assay to identify sequence contexts targeted by APOBEC3B and APOBEC3A. a** Schematic of the Oligo-seq assay. Created with BioRender.com. **b** Schematic of the 3-nt hairpin loop used to perform the Oligo-seq assay. N refers to any of the four DNA bases. A sequence logogram showing the fold enrichment and depletion for each of the four DNA bases at the −2 position after deamination of the cytosine by A3B (**c**) or A3A (**d**). **e** Schematic of the 4-nt hairpin loop used to perform the Oligo-seq assay. N refers to any of the four DNA bases. A sequence logogram showing the fold enrichment and depletion for each of the four DNA bases at the −2 position and −3 position after deamination of the cytosine by A3B (**f**) or A3A (**g**). **h** A river plot depicting the relative frequency of each nucleotide at the indicated position of a 4-nt hairpin loop and its association with the nucleotides located before or after. **i** Scatter plot of the fold enrichment and depletion for each of the 16 dinucleotide sequences possibly present in the 4-nt hairpin loop shown in (**e**). Source data are provided as a Source Data file.

or PCR amplification to the targets that are then purified using bead-based size-selection methods, which cannot be performed efficiently on short ssDNA. Therefore, we decided to adapt a methodology previously employed to sequence short single-stranded RNAs[72,73]. The first step of the previous method consisted of the ligation of a single-strand DNA adapter to the RNA using T4 RNA ligase. In Oligo-seq, we modified this step to allow the ligation between two ssDNAs. In order to facilitate the ligation process, we added 3′ and 5′ ssDNA tails on each side of the stem allowing the DNA binding of the ligase. We tested several DNA ligases and found that the ssDNA CircLigase efficiently ligated the target oligonucleotide with the single-stranded adapter (Fig. 2a, step 2 and Supplementary Fig. 2A). To best facilitate the DNA synthesis carried out by DNA polymerase as illustrated in step 3 of the library generation process (Fig. 2a) and prevent the DNA stem-loop from obstructing the polymerase, we opted for a 5-bp hairpin stem which was sufficient to block ExoT activity (Supplementary Fig. 2B). The 3-nt loop hairpin oligonucleotide was incubated with whole cell extract expressing either A3A or A3B. The extract was carefully titrated to a concentration that restricts the reactions to a maximum of 10% completion. It is crucial to conduct the reaction under these limiting conditions to favor deamination on the optimal substrates specifically. Importantly, we used whole-cell extracts depleted of UNG (Supplementary Fig. 2C), and recombinant UDG was not added to the reaction, in order to limit the conversion of uracil to abasic sites. During the library preparation, the cytosine deaminated by A3A or A3B (C-to-U deamination) is recognized as a T by the DNA polymerase leading to the conversion of the TpC motif to TpT (Fig. 2a, step 3). Deep sequencing was used to identify and separate sequences containing TpT events from TpC events. We then quantified the percentage of each nucleobase at the −2 position present in sequences containing TpT events compared to the total population (Fig. 2a, step 8). We found a strong enrichment of A and G (purine, R) on DNA substrates deaminated by A3B, whereas those targeted by A3A showed an enrichment of C or T (pyrimidine, Y) at that position (Fig. 2c, d). Therefore, these findings validate the efficacy of Oligo-seq as a reliable method for identifying the specific sequence contexts targeted by A3A and A3B, as it aligns with previous studies that demonstrate the respective preference of A3B and A3A for RTC and YTC motifs[44].

We next applied Oligo-seq on a second DNA substrate that forms a 4-nt loop hairpin with two randomized bases in positions −2 and −3 (Fig. 2e). Similar to the results obtained for a loop of 3 nt, A3B showed a preference for RTC and A3A for YTC (Fig. 2f, g). However, we found that the nucleotide at the −3 position also strongly impacts both A3B and A3A deaminase activity. Notably, we detected a clear preference for C in the case of A3B, while A3A exhibited a high preference for Y at this position (Fig. 2f, g). To better understand how the sequence context affects A3A and A3B activity, we conducted a second analysis focusing now on the dinucleotide motif preference. We deconvolved each deaminated sequence separately to determine the dinucleotide motif frequency relative to the total population and visualized them on river plot (Fig. 2h and Supplementary Fig. 2D). We then quantified the enrichment or depletion of all 16 possible sequence combinations that can arise from the two randomized bases. This deconvolution approach allowed us to systematically assess and characterize specific consecutive nucleotide sequence motifs favored by A3A or A3B (Fig. 2i and Supplementary Fig. 2E). This analysis further confirmed the strong preference of A3B for the 5′-CR motif and A3A for the 5′-YY motif preceding the TpC site. However, we found that the 5′-TR dinucleotide motif was also favored by A3B (Fig. 2i). This preference of A3B for a T in position −3 when A or G are in position −2 was initially masked in the single-nucleotide analysis by the low affinity of A3B for the 5′-TY motif, efficiently counterbalancing the enrichment for the 5′-TR motif. This observation stressed the importance of deconvolving each dinucleotide sequence separately to accurately reveal A3B's and A3A's preferred sequences. Taken together, these results revealed that A3B and

A3A preferentially target 4-nt hairpin loops with 5′-YR and 5′-YY motifs respectively.

## Biochemical analyses validate APOBEC3B and APOBEC3A substrates preferences

To validate the sequence preferences predicted by Oligo-seq and to examine the impact of the sequence context on A3B and A3A deaminase activity, we measured the catalytic activity of A3B and A3A on selected synthetic substrates using our in vitro assay. We first focused on 4-nt loop sequences found to be either preferentially targeted by A3B (5′-CA, and 5′-CG) or disregarded by A3B (5′-GA, and 5′-GC) (Fig. 2i). Titration of whole-cell extract expressing A3B showed a strong deamination preference for DNA stem-loop structures with 5′-CA, and 5′-CG motifs compared to 5′-GA, and 5′-GC motifs (Fig. 3a, b), confirming the results obtained with Oligo-seq. A3B exhibited around 70 times more deamination activity for DNA stem-loop with a 5′-CA motif compared to DNA stem-loop with a 5′-GA (Fig. 3B), demonstrating that a single nucleotide change within the loop can dramatically impact A3B activity. We further validated A3B preferences by testing other specific dinucleotide sequences favored or disfavored by A3B (Supplementary Fig. 3A) and demonstrated that A3B activity for hairpins with a 5′-CA motif was higher than for linear DNA (Supplementary Fig. 3B, C). Importantly, we verified that the measured deamination activity of the top targeted DNA stem-loop (5′-CG) was exclusively dependent on the presence of A3B in the cell extract (Supplementary Fig. 3D). Note that these results explain why we did not previously report a preference toward hairpin DNA for A3B. It has now come to light that the 4-nt loop hairpin with a 5′-GT sequence preceding the TpC motif, which was used in our previous study[15], proves to be a poor substrate for A3B compared to other sequences (Fig. 2f, h, i, and Supplementary Fig. 3A–C). This further stresses the importance of developing unbiased approaches such as Oligo-seq to study APOBEC substrate specificity. Finally, to eliminate any potential influence from competing proteins present in whole-cell extract that could affect A3B deamination of DNA stem-loops, we performed pulldown purification to isolate A3B from human cells expressing exogenous A3B fused to a FLAG tag as previously described in ref. 15. After pulldown purification from cell extract, A3B showed the same tendency to deaminate 4-nt hairpin loop with 5′-CA, and 5′-CG motifs rather than 5′-GC, and 5′-GA motifs or linear DNA (Supplementary Fig. 4A–D), ruling out the influence that other proteins present in the whole-cell extract may have on shaping A3B's substrate preference for specific sequences. Importantly, the presence of other cytosines in the loop of A3B's preferred hairpins did not impact the deamination level quantified. Given that the denaturing conditions of the electrophoresis provides resolution at the near nucleotide level, the cleavage resulting from deamination of these "off-target" cytosines can be detected at lower molecular weight (Supplementary Fig. 4E). Switching the preceding T to a C eliminates the secondary deamination product without affecting the deamination levels on the target TpC motif. (Supplementary Fig. 4F). Moreover, the position of the TpC motif within the loop is critical for A3B activity. When we moved the TpC site to the center of the loop, we almost completely abrogated A3B-induced cytosine deamination (Supplementary Fig. 4G). Together, these results demonstrate that A3B deaminase activity is regulated not only by the DNA secondary structure, but also by the position of the TpC site within the loop and the surrounding sequence.

We next selected top and bottom DNA stem-loops targeted by A3A from the Oligo-seq results (Fig. 2i). Note that the sequence 5′-TC was deliberately excluded as it would result in the formation of an additional deamination motif. The titration of whole-cell extract expressing A3A demonstrated a pronounced preference for deamination of stem-loop structures containing 5′-CT and 5′-CC motifs, while the presence of 5′-AG and 5′-GG motifs had a detrimental effect on A3A activity (Fig. 3c, d). A3A activity was about 15 times higher for DNA

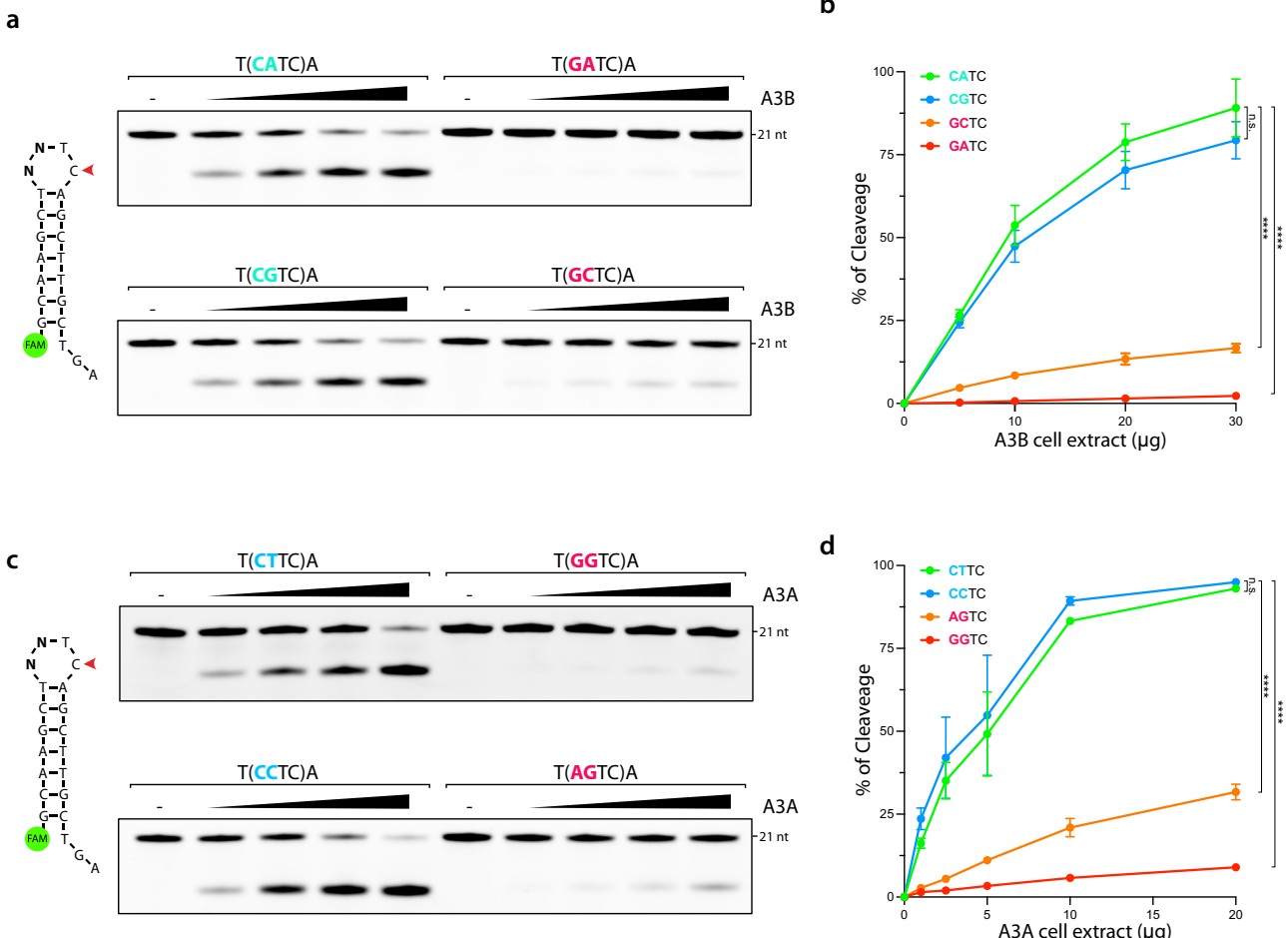

**Fig. 3 | Deamination of 4 nt hairpin loops by APOBEC3B and APOBEC3A is sequence context dependent. a** A3B deamination activity was performed with U2OS whole cell extract (5–30 μg) on 4-nt hairpin loops with the indicated loop sequences. The loop sequences are shown enclosed within parentheses. **b** Quantification of the A3B deamination activity shown in (**a**). Data are presented as mean values ± S.D. (Number of biological replicates, $n = 3$; n.s. not significant ($p = 0.1803$) and ****$p < 0.0001$ (two-tailed unpaired t-test)). **c** A3A deamination activity was performed with HEK-293T whole cell extract expressing A3A (1–20 μg) on 4-nt hairpin loops with the indicated loop sequences. The loop sequences are shown enclosed within parentheses. **d** Quantification of the A3A deamination activity shown in (**c**). Data are presented as mean values ± S.D. (Number of biological replicates, $n = 3$; n.s. not significant ($p = 0.0942$) and ****$p < 0.0001$ (two-tailed unpaired t-test)). Source data are provided as a Source Data file.

hairpins with 5'-CT compared to the 5'-GA sequence (Fig. 3d). Moreover, we confirmed that the deamination monitored on the DNA stem-loop with 5'-CT was exclusively dependent on the presence of A3A in whole-cell extract (Supplementary Fig. 4H), and when purified by immunoprecipitation from cell extract, A3A revealed identical substrate preference (Supplementary Fig. 4B, I). Taken together, these results demonstrate that the sequence preceding the TpC site in the context of a hairpin strongly impacts both A3B and A3A deaminase activity. Furthermore, A3B and A3A exhibit distinct sequence preferences, suggesting the generation of differential mutational landscapes in cancer associated with A3B and A3A.

**Comprehensive analysis of APOBEC3B's substrate preferences**
We then applied a similar analysis to identify the sequence context on hairpins with longer loops. Comparison of Oligo-seq results obtained from A3B and A3A deamination revealed a robust preference by A3B for 5-nt-loop structures containing a 5'-CCR motif (Fig. 4a, b and Supplementary Fig. 5A, B). On the other hand, we found that the presence of an A or G nucleotide at the −4 position had a detrimental effect on deamination efficiency for both A3A and A3B (Fig. 4a and Supplementary Fig. 5B). Deconvolution of each trinucleotide sequence further highlighted the depletion of stem-loop sequences with a G in

−4 position (highlighted in purple) (Fig. 4c and Supplementary Fig. 5C, D). Within a 5-nt loop, the presence of a G nucleotide at the −4 position results in base pairing with the C nucleotide at position 0, causing the loop to shrink to 3 nt. This conformational change makes the cytosine residue inaccessible for deamination by both A3A and A3B, thereby explaining the observed poor deamination of these targets (Fig. 4d). We therefore selected two DNA stem-loop sequences depleted in the Oligo-seq experiment that still maintain a 5-nt loop structure (5'-AAC and 5'-AAT) (Fig. 4c). Consistently, A3B demonstrated low deamination activity for these substrates, which is only slightly higher than the substrates with a 3-nt loop that masks the cytosine as predicted from the Oligo-seq experiment (Fig. 4d, e).

We next conducted Oligo-seq on hairpins featuring a 6-nt loop that was also favored by A3B (Fig. 1e) and identified 5'-CCGR as the preferred sequence context (Supplementary Fig. 6A–C). Similar to the 5-nt loop, the presence of G nucleotide in −5 position led to the formation of a 4-nt loop through base pairing with the C in position 0, inhibiting A3B activity (Supplementary Fig. 6B, C). However, upon comparing hairpins that maintain a 6-nt loop, we observed a clear preference of A3B for 5'-CCGR motifs, while 5'-AAAY motifs were found to negatively impact A3B activity (Supplementary Fig. 6C, D). Note that similar to the loop of 4 nt, we switched the T:A base pair closing the

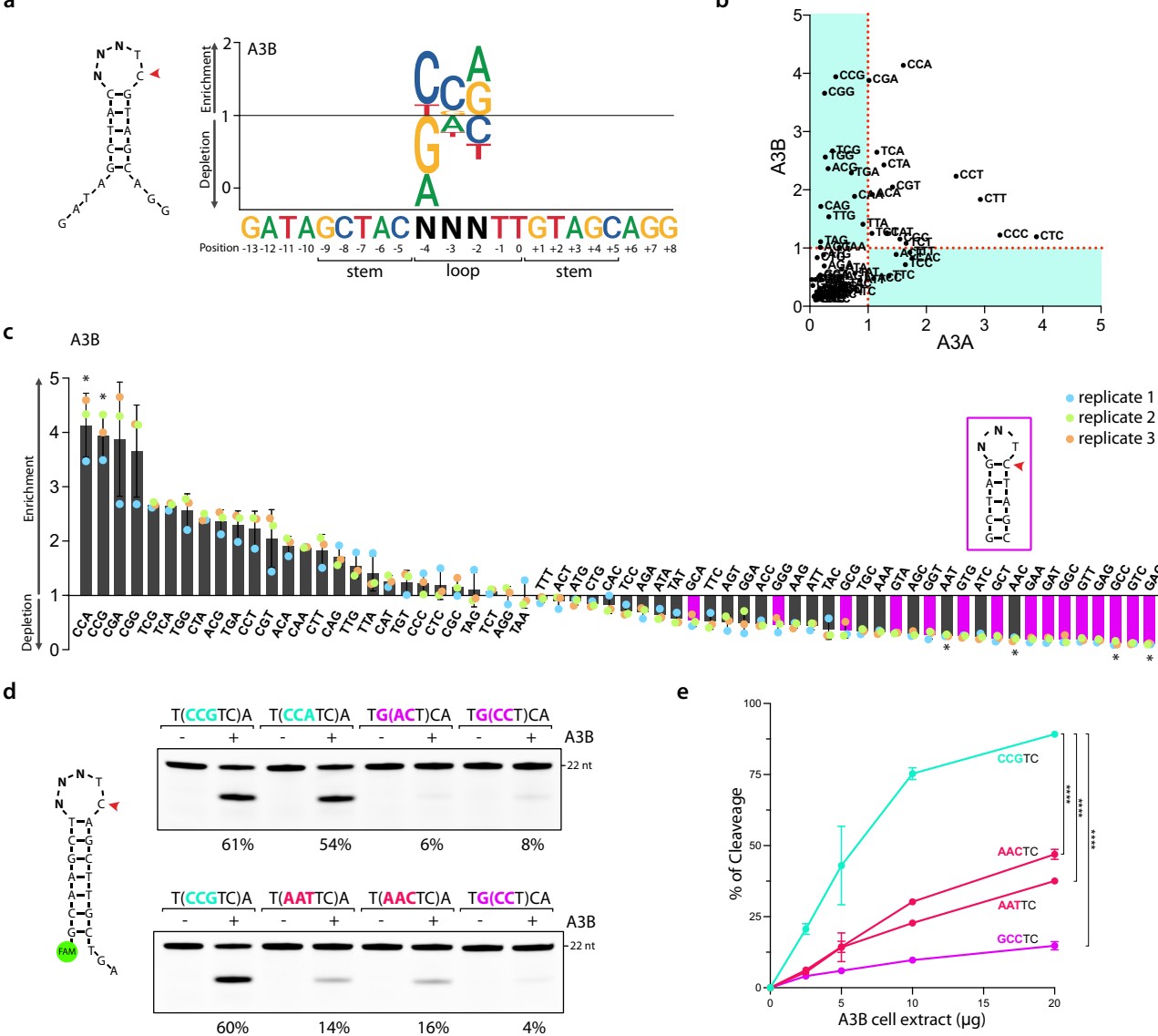

**Fig. 4 | Identification and validation of 5-nt hairpin loop sequences preferentially targeted by APOBEC3B. a** A sequence logogram showing the fold enrichment and depletion for each of the four DNA bases at the −2, −3, and −4 position after deamination of the cytosine by A3B. **b** Scatter plot of the fold enrichment and depletion for each of the 64 trinucleotide sequences possibly present in the 5-nt hairpin loop shown in (**a**). **c** Bar graph showing the fold enrichment and depletion for each of the 64 trinucleotide sequences possibly present in the 5-nt hairpin loop shown in (**a**) after deamination by A3B. The purple-colored bars indicate 5 nt hairpin loop with sequences that form 3-nt hairpin loops. Asterisk (*) indicates the sequences validated in vitro (**d**). Data are presented as

mean values ± S.D. (Number of biological replicates, $n = 3$). **d** A3B deamination activity assay was performed with 20 µg of U2OS whole cell extract expressing endogenous A3B, on indicated DNA stem-loop oligonucleotides. The loop sequences are shown enclosed within parentheses. The percentage of cleavage is indicated. **e** Quantification of the A3B deamination activity on the indicated oligonucleotides. The deamination assay was performed with U2OS whole cell extract (2.5–20 µg). Data are presented as mean values ± S.D. (Number of biological replicates, $n = 3$; ****$p < 0.0001$ (two-tailed unpaired t-test)). Source data are provided as a Source Data file.

stem of the hairpin with a C:G base pair to eliminate potential off-target caused by the presence of another TpC motif. This alteration in the stem sequence did not significantly impact A3B activity (Supplementary Fig. 6E).

Finally, we performed Oligo-seq on linear DNA. We noticed a strong enrichment for G nucleotides in +1 position after both A3B and A3A deamination (Supplementary Fig. 7A, B). These results are consistent with a previous study showing A3A preference for a G following the TpC site[40]. In agreement, we have demonstrated that the substitution of a G with a T at the +1 position reduces the activity of A3B deaminase (Supplementary Fig. 7C). Altogether, these findings further underscore the significance of the sequence context surrounding the TpC motif in governing A3B deaminase activity.

## APOBEC3B and APOBEC3A substrate preferences are dictated by loop 1

After having identified A3B's substrate preferences, we reconducted a comparison of A3B's deamination activity across hairpins containing the optimal sequences. We found that A3B still preferentially deaminated DNA stem-loop structures with a loop size of 4 to 6-nt rather than 3-nt or linear DNA (Fig. 5a). We next compared A3B and A3A substrate selectivity on their respective preferred hairpin DNA. We selected a hairpin with a loop of 3 nt with a TTC sequence known to be preferentially targeted by A3A[15] and compared it to a 5-nt loop hairpin loop that we found to be highly deaminated by A3B. A3B displayed a substantial 20-fold increase in deamination activity for its preferred hairpin of 5 nt compared to the 3-nt stem-loop (Fig. 5b). On the

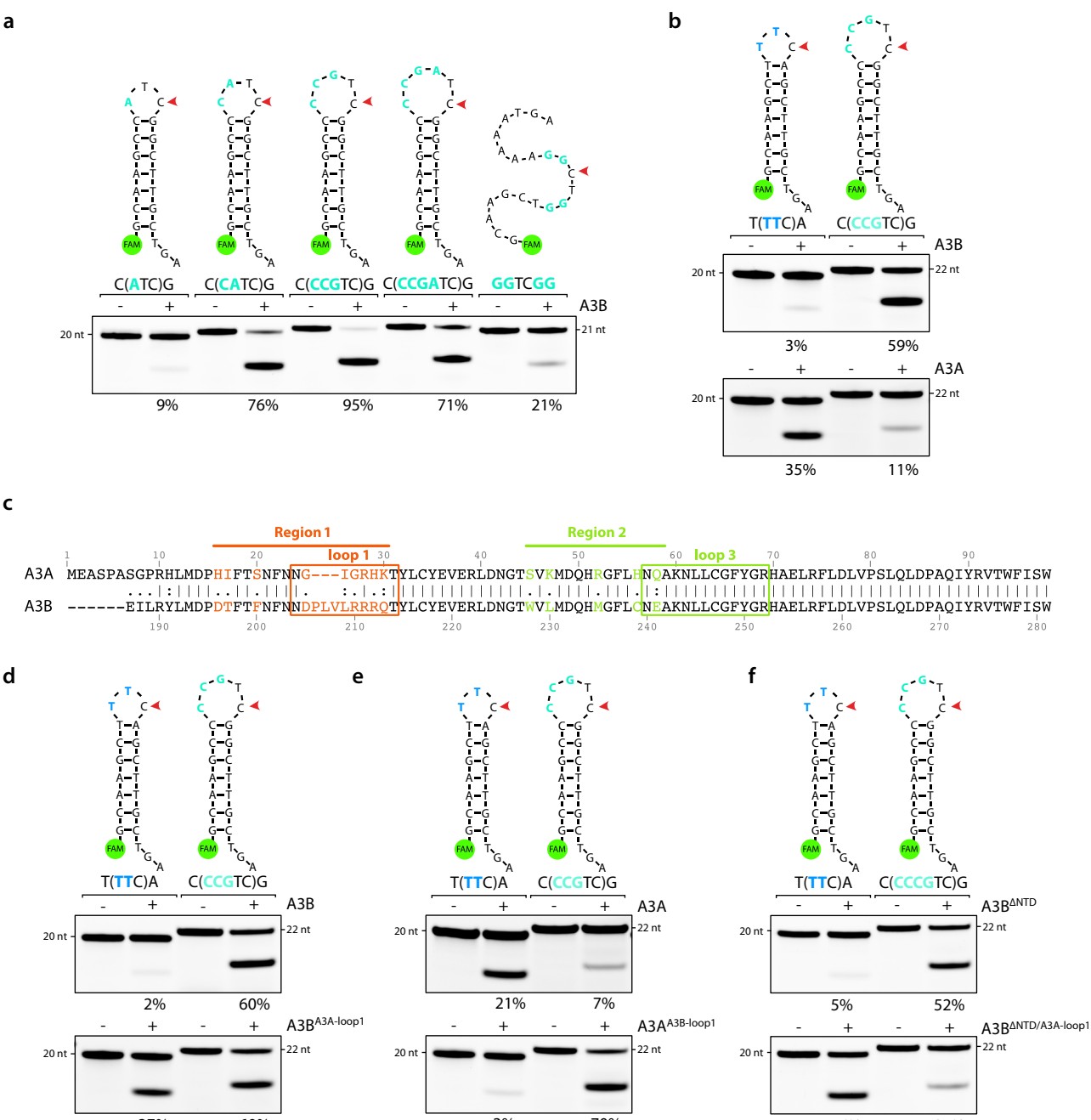

**Fig. 5 | Loop 1 of APOBEC3B and APOBEC3A defines their DNA substrate preference. a** A3B deamination activity was performed with 20 μg of U2OS whole cell extract on the indicated substrates. The loop sequences are shown enclosed within parentheses. The percentage of cleavage is indicated. **b** Deamination activity assay was performed with 20 μg of U2OS whole cell extract expressing A3B or 5 μg of HEK-293T whole cell extract expressing A3A on a 3-nt and 5-nt DNA stem-loop oligonucleotides preferentially targeted by A3A and A3B respectively. The loop sequences are shown enclosed within parentheses. The percentage of cleavage is indicated. **c** Alignment of the amino acid sequences for A3A (amino acid 1–98) with the C-terminus of A3B (amino acid 186 to 281). The amino acids that form loop 1 and loop 3 are enclosed in boxes. The highlighted color-coded residues indicate the amino acids that differ between A3B and A3A. The residue numbers at the top and bottom correspond to those found in the wild-type full-length A3A and A3B proteins. **d**–**f**. Deamination activity assay using the indicated constructs expressed in HEK-293T cells was monitored on both A3A's and A3B's preferred DNA stem-loop respectively. The percentage of cleavage is indicated. Source data are provided as a Source Data file.

contrary, A3A exhibited clear disfavor for A3B's preferred substrates, demonstrating a 3-fold increase in deamination activity towards the 3-nt loop hairpin, as compared to the 5-nt loop hairpin (Fig. 5b). Altogether, we established that both A3B and A3A preferentially target distinct types of DNA substrates, which differ by both their secondary structures and their sequence contexts.

We then aimed to pinpoint specific A3B and A3A structural features that dictate their substrate preferences. We first asked whether

the differing amino acids between the two regions surrounding loop 1 and loop 3 of both A3B$^{CTD}$ and A3A may play a pivotal role in determining A3B's substrate preference given their proximity to the active site pocket (Fig. 5c). We replaced the amino acids of these regions in A3B with those from A3A. We found that A3B$^{A3A-Region1}$ increased A3B substrate preferences for the 3-nt hairpin loop, whereas A3B$^{A3A-Region2}$ did not affect A3B substrate selectivity (Supplementary Fig. 8A). We then switched A3B loop 1 (DPLVLRRRQ) with A3A loop 1 (GIGRHK) and

vice versa. Similar to A3B[A3A-Region1], A3B[A3A-loop1] showed an increase in deaminase activity toward the 3-nt hairpin loop compared to A3B wild-type (Fig. 5d). In contrast, A3A[A3B-loop1] preferentially deaminated 5-nt hairpin loops and lost almost all deamination preference for the 3-nt hairpin loop targeted by A3A wild-type (Fig. 5e). To understand why A3B[A3A-loop1] still showed significant activity for both types of hairpin loops, we next focused on the A3B's NTD as another key structural difference compared to A3A. The deletion of the NTD (A3B[ΔNTD]) did not affect A3B's preference for 5-nt versus 3-nt hairpin loops (Fig. 5f), demonstrating that the NTD has no impact on A3B substrate selectivity. However, the fusion of A3B's NTD with the N-terminal of A3A generated a chimera protein (A3B[NTD]-A3A) that mirrored the activity observed for A3B[A3A-loop1], whereas A3B[NTD]-A3A[A3B-loop1] replicated A3B wild-type substrate preference (Supplementary Fig. 8B). On the other hand, the deletion of the NTD from A3B[A3A-loop1] (A3B[ΔNTD/A3A-loop1]) exhibited substrate preference like wild-type A3A (Fig. 5e, f), demonstrating the necessity of modifying the loop 1 and removing the NTD domain of A3B to mirror A3A's substrate preferences. Taken together, these results demonstrate that loop 1 is the critical structural feature shaping A3B and A3A substrate selectivity.

## APOBEC3B induces mutations in DNA stem-loop structures of tumor genomes

The ability of A3B to deaminate TpC sites in specific DNA stem-loops in vitro prompted us to investigate whether mutations accumulate at hairpin-forming sequences in tumor genomes. A significant challenge in evaluating the overall impact of A3B deaminase activity on the mutational landscape of human tumor cells is the presence of A3A attributable mutations in many tumors, which can mask those generated by A3B[6,15]. To overcome this challenge, we first conducted an analysis of whole genome sequencing (WGS) data obtained from mouse tumors caused by the expression of human A3B[74]. Due to the absence of a direct equivalent of the human *A3A* and *A3B* genes in mice[51], any A3B-induced mutations in the mouse genome will not be confounded by mutations generated by A3A. We next aggregated mutation statistics across disparate genomic sites with the potential to form hairpins with a 3- to 6-nt loop from these WGS dataset. We observed a higher mutation rate in hairpins with a loop of 4 nt and a TpC site at the 3' side of the loop (termed "A3B optimal hairpins") (Fig. 6a and Supplementary Data 1). Furthermore, we found that the frequency of mutations in 4 nt hairpins loop increased with high stem strength (here defined as #AT basepairs + 3 × #GC basepairs[15,16]) (Supplementary Fig. 9A). In DNA stem-loops with the strongest pairing, the mutation frequency increased up to 7-fold when the TpC site was located on the 3' side of the loop compared to other positions. In contrast, the mutation frequency remained unchanged for other positions within the loop (Fig. 6a and Supplementary Fig. 9A). This finding further suggests that the precise positioning of TpC residues plays a critical role in facilitating optimal A3B deaminase activity.

Among the mouse tumors expressing A3B, we identified a total of 32 mutations in A3B optimal hairpins with a stem strength of 12 or higher. Remarkably, the deconvolution analysis of each dinucleotide sequence preceding the TpC residues revealed a marked prevalence of mutations in the best sequences motifs targeted by A3B that paralleled motif preferences identified by oligo-seq and validated using the in vitro deaminase assay (Supplementary Fig. 9B). Additionally, it should be noted that we also found significant levels of mutations in 3 nt loop with TpC motifs at the 3'-most position but not in 5-nt or 6-nt hairpin loops (Fig. 6a). These observations suggest that mutations mediated by A3B in cancer genomes may be influenced not just by the substrate selectivity of A3B, but also by the cells' ability to form hairpins. Indeed, longer loop lengths can negatively impact the stability of the structure[75,76]. In addition, the presence of a longer ssDNA may facilitate the recruitment of DNA helicases or

other proteins, resulting in the dissociation of DNA stem-loops that are known to be sources of genomic instability for the cells[77,78]. The observed prevalence of mutations in 3- and 4-nt hairpin loops could be attributed to the balance between A3B's preferred substrates and the higher probability of smaller hairpin loops forming within cells. Therefore, we propose that the mutational landscape caused by A3B may be determined by both A3B's substrate selectivity and potential cellular mechanisms that actively inhibit the formation of such structures.

To further delineate the different types of hairpin mutations generated by A3B and A3A, we conducted a parallel analysis on mouse tumors driven by A3A expression[79]. In agreement with our previous studies[15,16], we found that A3A-induced mutations preferentially occur at genomic sites that form hairpins with a 3 nt loop and a TpC site located at the 3' end (termed "A3A optimal hairpins") (Fig. 6b). Ultimately, these results suggest that A3A and A3B generate a distinct mutation landscape in cancer genomes, driven by their unique substrate specificity.

## APOBEC3A and APOBEC3B mutation landscape in human tumors

We next investigated whether A3B-induced DNA stem-loop mutations can be detected in human tumors. Based on the analysis performed in our previous studies[15,16,18,28], we examined WGS data of 2644 tumors of multiple cancer types to identify APOBEC mutations in tumors that were driven by A3A or A3B. To achieve this, we classified mutations in patient tumors by (1) tumor type, (2) frequency of APOBEC-signature mutations, and (3) enrichment for APOBEC mutations in A3A-preferred YTC motif or A3B-preferred RTC motif (Fig. 6c). This analysis resulted in a Y-shaped or "bird plot" that separates A3A-dominated tumors (right wing) from A3B-dominated tumors (left wing). We next selected tumor samples with the most A3A (A3A+) and most A3B (A3B +)-induced mutations (outlined in red or in blue respectively), and monitored the levels of mutated hairpins in these specific tumor samples. Note that from our selection, we excluded patient tumors that contain 10% of their mutations assigned to MSI, Smoking, UV, POLE, or ESO mutational signatures to avoid any masking effects on APOBEC-induced mutations. We found that A3B+ patient tumors manifested a strong prevalence for mutations in A3B optimal hairpins (Fig. 6d) and are enriched for motifs targeted by A3B (Supplementary Fig. 9B). Conversely, A3A+ tumors exhibited higher mutation rates in A3A optimal hairpins (Fig. 6e). It is also important to highlight the striking similarity of the mutation patterns caused by A3A or A3B observed between mouse and human datasets (A3B mouse tumors versus A3B human tumors: Pearson correlation 0.7127 [*p*-value 0.000924] and A3A mouse tumors versus A3A human tumors: Pearson correlation 0.9795 [*p*-value 1.482 × 10^{-12}]) strengthening the robustness of the results (Fig. 6a, b and d, e). Moreover, we found higher levels of mutated hairpin DNA in tumors driven by A3A compared to tumors associated with A3B mutations. This result corroborates earlier findings that suggest A3A is the primary driver of the APOBEC mutational signature in cancer[6,15,17,18,28,68].

Lastly, we conducted an analysis in each individual tumor sample to identify those dominated by A3B-induced hairpin mutations over A3A-induced hairpin mutations (described as A3B hairpin mutation character), and vice versa (A3A hairpin mutation character). To determine A3A or A3B mutation characters, we calculated the ratio between the levels of 4-nt hairpin loop and 3-nt hairpin loop mutated by A3B and A3A respectively, multiplied by the ratio of mutated ATC versus TTC sites (motifs targeted by A3B and A3A respectively) present in the 4-nt hairpin loops with TpC motifs positioned at the 3' end. Remarkably, A3A-dominated patient tumors showed a strong accumulation of A3A hairpin mutation character (red colored dots), whereas A3B-dominated patient tumors were predominantly enriched for A3B hairpin mutation character (blue colored dots) (Fig. 6f). More

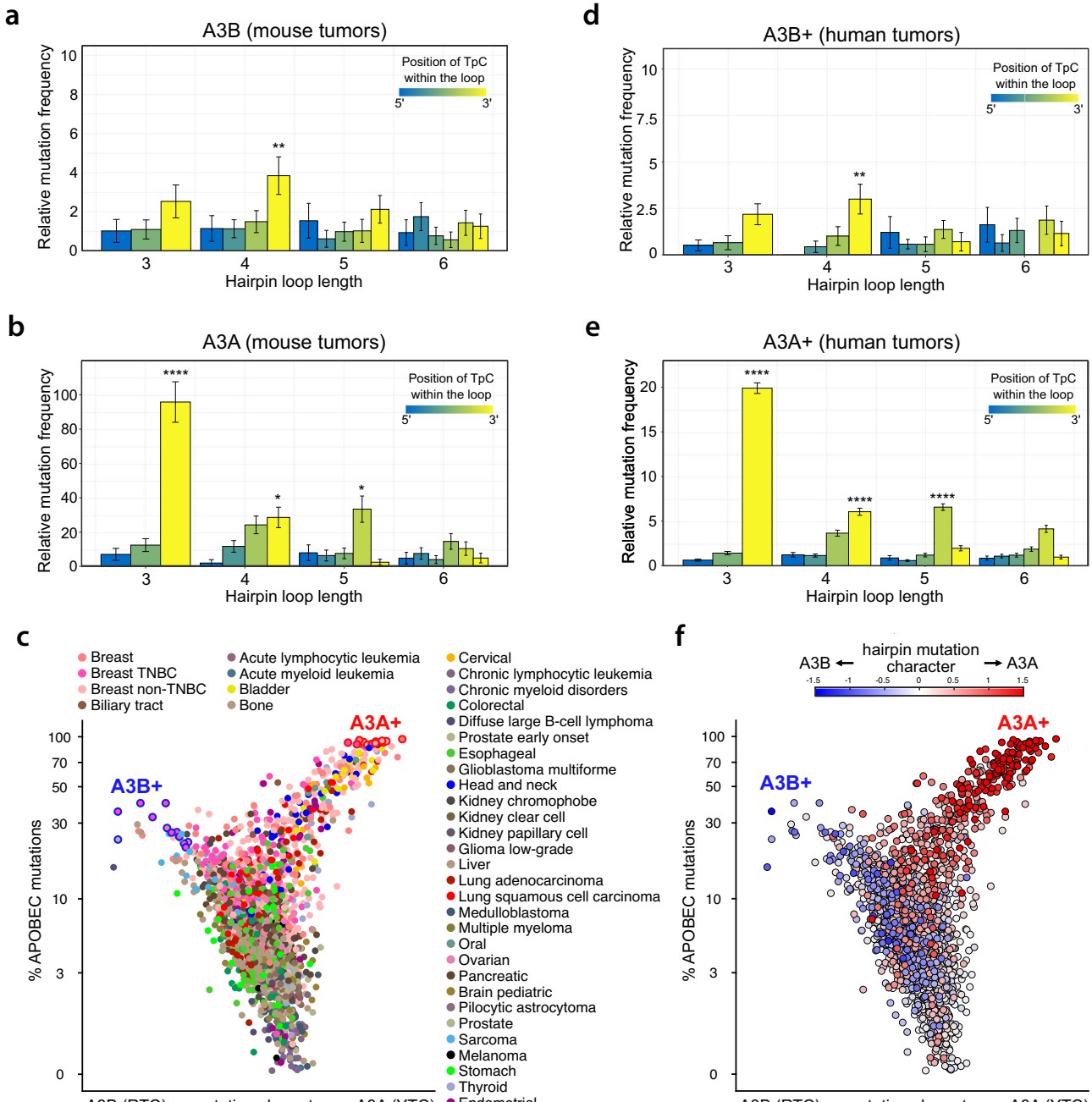

**Fig. 6 | APOBEC3B promotes mutations in hairpin-forming sequences in mouse and human tumors.** Relative mutation frequency in mouse tumor samples expressing A3B (**a**) or A3A (**b**). The relative mutation frequency was calculated as the ratio of the number of mutations to the number of available sites, and normalized so that the background mutations rate (at all C:G positions) is equal to 1. Mutation levels were classified in terms of the size of hairpin loops (x-axis) and by their position in the hairpin loop (color gradient). Error bars represent 95% confidence intervals. *p*-values were calculated by performing a binomial test for each group and we adjusted these *p*-values for multiple comparisons using the Benjamini & Hochberg method (**p* = 4.93 × 10⁻³ for (**a**) and ****p* = 2.62 × 10⁻³¹, *p* = 4.11 × 10⁻² and *p* = 2.48 × 10⁻² for (**b**) [one-sided]). **c** Whole Genome Sequencing (WGS) of patient tumor samples were analyzed for their mutation frequency in the

TpC motif. Each patient's tumor samples were plotted by their level of mutations in the TpC motif and their mutation frequency in RTC versus YTC sequences. Samples with more than 500 mutations and with a fraction of POLE signature of less than 5% were sectioned (2644 tumors samples out of 3004). Dots were color-coded by tumor types. Relative mutation frequency in human tumor samples dominated by A3B mutation (**d**) or A3A mutations (**e**). Mutation levels were classified in terms of the size of hairpin loops (x-axis) and by their position in the hairpin loop (color gradient). Error bars represent 95% confidence intervals (**p* = 7.10 × 10⁻³ for (**d**) and ****p* = 0, *p* = 1.79 × 10⁻¹⁰, and *p* = 5.85 × 10⁻¹⁹ for (**e**) (one-sided)). **f** Patient tumor samples were plotted as described in (**c**). Dots were color-coded as a function of the A3A hairpin character (red colored dots) or A3A hairpin character (blue colored dots). Source data are provided as a Source Data file.

importantly, this approach enabled us to increase the resolution needed to distinguish patient tumors driven by A3A or A3B, especially for tumors that were present in the body of the bird plot and not assigned to either an A3A or A3B-dominated category due to the lower levels of APOBEC mutations and lack of significant differences between RTC

and YTC mutation rates (Fig. 6f and Supplementary Fig. 9C). Taken together, the mutation analysis of both mouse and human tumors highlights how A3A and A3B can each generate distinct mutation landscapes in cancer genomes, driven by their unique substrate selectivity.

## Discussion

Mutations induced by APOBEC enzymes represent one of the most prevalent mutation signatures found in cancer[6,11,19]. These mutations contribute significantly to tumor heterogeneity, facilitate metastasis, and play a crucial role in the development of drug resistance mechanisms[28,31,32,37,80]. A3A and A3B are the two APOBEC members mainly responsible for the APOBEC signatures detected in tumors[6–10,17,27]. However, it has been unclear whether A3A and A3B contribute to similar or distinct mutational landscapes in cancer genomes. Previous studies revealed that A3A has a strong preference for pyrimidines before the TpC motif, whereas A3B favors purines[44]. Moreover, A3A preferentially targets specific DNA stem-loop structures[15,16,18,45,49,68]. Nevertheless, any substrate preferences of A3B beyond its affinity for a purine preceding the TpC motif remained to be fully established. In this study, we demonstrated that A3B also selectively targets hairpins but that are different in terms of structures and sequence contexts from those targeted by A3A. We identified structural features on A3A and A3B responsible for their divergent substrate selectivity. Moreover, we determined that A3A and A3B induces mutations in different types of DNA stem-loops in cancer genomes. Consistent with the findings of our study, an accompanying paper from Dr. Ashok Bhagwat and his group employed an *Escherichia coli*-based system that expresses A3B. They observed comparable mutation patterns in hairpins further supporting our findings in mouse and human tumors[81]. Finally, by leveraging the distinct mesoscale preferences of both A3A and A3B, we successfully differentiated patients' tumors that accumulate an APOBEC mutational signature driven be either A3A, A3B, or both, laying foundations for exploring the roles of A3A and A3B in tumor evolution and drug resistance.

Enzymes possess inherent biochemical properties that define their functionality, notably catalytic activity, processivity, and selectivity. Catalytic activity denotes the efficiency of an enzyme in catalyzing a specific biochemical reaction. Processivity, on the other hand, refers to the enzyme's ability to perform multiple catalytic cycles without dissociating from its substrate or template. Lastly, selectivity represents the enzyme's capacity to discern and differentiate between various substrates. Previous studies have characterized structural features of A3A and A3B for regulating their catalytic activity and processivity, but less is known about which domains modulate their substrate selectivity. Comparative biochemical studies between A3A and A3B demonstrated that A3A has a greater catalytic activity level. This difference in activity was attributed to the differential amino acid sequence found in A3A and A3B's loop 1[41,56]. Mutation of the aspartate 131 in loop 7 to a glutamate switched A3A substrate selectivity toward CpC motifs[40]. Moreover, the N-terminal domain of A3B was shown to be critical in promoting both A3B catalytic activity and processivity by promoting ssDNA interaction and self-interaction[52,54,55]. In this study, we revealed that A3A and A3B loop 1 not only determines the catalytic activity, but also plays a crucial role in regulating substrate selectivity, whereas the A3B NTD domain has no effect on A3B's substrate preferences. Consistently, the switch of A3A loop 1 with A3B loop 1 was sufficient to fully restore the preference for larger hairpin loops of specific sequence context.

A3B adopts a closed conformation due to the interaction between loops 1 and 7, specifically through residues Arg210-211-212 in loop 1 and Tyr315 in loop 7[41,56,82]. Therefore, A3B's ability to bind to ssDNA requires several conformation changes to break these interactions and to facilitate ssDNA entry into the active site pocket[56]. It is possible that specific DNA secondary structures and sequences preceding the TpC motif are critical to switch the active site to an open conformation. In contrast, A3A does not require a conformational switch, which provides one explanation for its higher activity. Furthermore, the higher deaminase activity of A3A is mediated in part by His29 which makes dual phosphate contacts to clamp the cytosine down in the active site[56–58,82]. The absence of His29-equivalent in A3B's loop 1 might

reduce A3B's ability to lock the cytosine in the active site of hairpins with a tight U-turn. In addition, Oligo-seq results revealed the importance of having pyrimidine nucleotides preceding YTC or RTC motifs to promote A3A or A3B activity respectively. The conserved pyrimidine preference by both A3A and A3B may be owed to the fact that pyrimidine bases are smaller than purines and have C=O to form H-bond interaction. A recent structural study detailing A3A's interaction with hairpin DNA revealed that the His29 base-stacks with the nucleotide at +1 causing the pyrimidine at −2 to base-stack with the pyrimidine at position −3 and stabilizing the tight turn of the DNA[50]. This observation is consistent with A3A's preferred 5′-YYTC sequence we identified in a hairpin with a 4-nt loop. However, future studies focusing on determining the exact structure of the complex between wild-type A3B and a linear DNA or a DNA stem-loop will be crucial to gain a better understanding of A3B's structural basis for selecting specific types of DNA structures.

APOBEC-signature mutations are preferentially enriched on the lagging-strand template of DNA replication forks[20,22,83], suggesting that transiently exposed ssDNA during replication might be the source of the hairpin DNA structures targeted by A3A and A3B. In addition to being susceptible to mutations, the formation of DNA stem-loops in cells can contribute to genomic instability, resulting in replication fork stall or collapse, and DNA double-strand breaks[77,78]. For example, the MRX complex has been found to cleave hairpins that form during the synthesis of the lagging strand in yeast, which leads to DNA double-strand breaks[84,85]. Hence, it is crucial for cells to prevent or counteract the formation of hairpins. To achieve this, cells employ the ssDNA binding protein RPA along with DNA helicases such as BLM and WRN to actively suppress the occurrence of DNA stem-loops[86–88]. Thus, it is tempting to speculate that hairpins with smaller loops which form more stable thermodynamic structures[75,76], confer enhanced resistance against cellular mechanisms that suppress their formation. Consequently, it would increase the likelihood of mutations by A3A or A3B in DNA stem-loop structures that are more stable rather than the more optimal. The balance between A3B's preference for hairpins with longer loops and a cell's potential to have more hairpins with small loops could provide an explanation for the lower A3B-associated mutation levels in hairpins detected in tumor genomes compared to the high frequency of hairpin mutations induced by A3A[15,16,18]. Moreover, because A3B is a less active enzyme than A3A[40,56], hairpins stability might more strongly influence the total level of A3B-induced mutations detected in hairpins. Nevertheless, future studies are necessary to elucidate the impact of the different cellular mechanisms that suppress hairpin formation in APOBEC-induced mutations in tumors.

Recent development of base editing technologies has resulted in numerous chimeric enzymes that fuse deaminase enzymes to catalytically impaired Cas9 protein to correct genetic diseases by generating specific mutations in genomic DNA[63,64]. However, the efficiency of these base editors relies on their ability to interact and deaminate the target DNA[65]. The sequence and secondary structure surrounding the target sites can strongly impact the deaminase activity of the base editors. Thus, the selection of the best enzymes to deaminate a specific site is critical for successful correction of a mutation by base editing. The use of Oligo-seq on synthetic substrates can provide a simple method for predicting the efficacy of various base editors on a specific target, enabling the selection of the most suitable one. Moreover, ongoing efforts to evolve new cytosine or adenosine base editors with improved efficiency and targeting capacities on different sequence contexts can greatly benefit from the application of Oligo-seq. Indeed, Oligo-seq can quickly assess how each modification affects the sequence and secondary structure recognition of the deaminase, thereby facilitating the optimization of these new base editors.

Beyond base editors and deaminase enzymes, Oligo-seq holds significant potential for investigating mesoscale preferences of other

DNA-modifying enzymes, including DNA methylation enzymes and repair factors that recognize specific DNA modifications induced by various stresses (such as deaminated bases or oxidation of guanine [8-oxoguanine]). DNA repair pathways, particularly those involved in error-free repair such as the base excision repair (BER) pathway, may be impacted by the mesoscale features surrounding the deaminated cytosine. These surrounding features can potentially decrease the accurate and efficient repair process, thereby promoting the formation of mutations. Indeed, it has been demonstrated that the depletion of specific DNA repair factors increases the APOBEC mutational signature or modifies the types of mutations generated by A3A or A3B[6,89,90]. Therefore, Oligo-seq can be adapted to study how BER factors, e.g., DNA glycosylases or APEX1 (DNA-apurinic or apyrimidinic site endonuclease), are affected by the presence of secondary structures and DNA sequence contexts. The importance of better determining how mesoscale genomic features influence the activities of these enzymes is crucial for unveiling the intricate interplay between the local genomic environment, the regulation of diverse cellular functions, and the subsequent consequences on genomic stability.

## Methods

### Plasmids

APOBEC3A and APOBEC3B cDNAs were synthesized by GenScript with a beta-globin intron and a Flag tag in C-terminus. The plasmids expressing APOBEC3A-GFP/Flag, APOBEC3B-GFP/Flag, and APOBEC3B-CTD-GFP/Flag (amino acids 187–382) were generated by inserting the cDNA into the pcDNA-DEST53 vector using the Gateway Cloning System (Thermo Fisher Scientific)[34]. pcDNA 3.1(+)-A3B$^{A3A-Region1}$-Flag, pcDNA 3.1(+)-A3B$^{A3A-Region2}$-Flag, and pcDNA 3.1(+)-A3B$^{A3A-loop1}$-Flag, and pcDNA 3.1(+)-A3B$^{A3B-loop1}$-Flag were generated using site-directed mutagenesis PCR on a pcDNA 3.1(+)-A3B-Flag construct previously described in refs. [47,52]. All the other APOBEC3B or APOBEC3A mutants were constructed by site-directed mutagenesis using pcDNA-DEST53-A3B-Flag as a backbone. For A3B$^{NTD}$-A3A, the amino acids 1-186 of A3B were fused with amino acids 7 to 199 of A3A by site-directed mutagenesis using pcDNA-DEST53-A3B-Flag as a backbone.

### Cell culture

U2OS and HEK-293T cells were maintained in Dulbecco's modified Eagle's medium (DMEM) supplemented with 10% fetal bovine serum (FBS) and 1% penicillin/streptomycin.

### RNA interference

siRNA transfections (4 nM) were performed by reverse transfection with Lipofectamine RNAiMax (Thermo Fisher Scientific, #13778075). siRNAs were purchased from Thermo Fisher Scientific (Silencer Select siRNA). The sequences of the siRNAs used in this study were:

siCTL: Catalog #4390846
siUNG1/2 (s14678): 5'-GCAUUACACUGUUUAUCCAtt

### CRISPR-Cas9 knockout cells

A3B knockout cell lines were generated as described in ref. [91] by transfection of the pSpCas9(BB)−2A-Puro (PX459) plasmid containing A3B gRNAs with FuGENE 6 Transfection Reagent (Promega, #E2691). 16 h after transfection, cells were selected with puromycin (1 µg/ml) for 2 days. For every target, three or more independent clones were generated. gRNA sequences used in this study were:

sgA3B#1GGCGGGCGGCCAGAGATGGTC
sgA3B#2GCGGGCGGCCAGAGATGGTCA

### Antibodies

The antibodies used in this study were: GAPDH polyclonal antibody (EMD Millipore #ABS16, 1/20000), Flag polyclonal antibody (Sigma-Aldrich, M2, #F7425, 1/3000), Flag monoclonal antibody (Sigma-Aldrich #F1804, 1/1000), UNG polyclonal antibody (Novus biologicals # NBP1-49985, 1/1500), RPA32 monoclonal antibody (Invitrogen, 9H8, #MA1-26418,1/2000), Vinculin monoclonal antibody (Sigma-Aldrich, hVin-1, #V9264, 1/5000), and APOBEC3B monoclonal antibody (5210-87-13; NIH-ARP #12398, 1/1000)[92].

### Cell lysate preparation

Whole cell extracts were prepared as described in refs. [15,16]. The APOBEC deamination assays were performed with cell extracts derived from either U2OS expressing endogenous levels of A3B or HEK-293T cells transiently expressing A3A, A3B, and derived mutants tagged with Flag and GFP. Cells were lysed in 25 mM HEPES (pH 7.9), 10% glycerol, 150 mM NaCl, 0.5% Triton X-100, 1 mM EDTA, 1 mM MgCl$_2$, and 1 mM ZnCl$_2$, and protease inhibitors. Cell lysates were sonicated three times for 10 s (50% output) and centrifuged 5 min at 20,000 × $g$ at 4 °C to remove the insoluble fraction. Then, 0.2 mg/mL of RNase A was added to the cell extract and incubated for 20 min at 4 °C. The additional insoluble fraction was removed by centrifugation for 10 min at 20,000 × $g$ at 4 °C. Protein concentration of the supernatant was determined by Bradford assay (Bio-Rad), and stored at −80 °C.

### DNA deaminase activity assay

The deamination assays were performed as described in refs. [15,16]. Reactions (50 µL) containing 8 µL of a normalized amount of cell extracts (expressing A3A, A3B, or indicated mutants) were incubated at 37 °C for 1 h in a reaction buffer (42 µL) containing a DNA oligonucleotide [20 pmol of DNA oligonucleotide, 50 mM Tris-HCl (pH 7.5), 1.5 units of uracil DNA glycosylase (New England BioLabs, #M0280), and 10 mM EDTA]. Then, 0.5 µL of 10 M NaOH was added to the reaction followed by 40 min incubation at 95 °C. Formamide was added to the reaction (50% final) and the reaction was incubated at 95 °C for 10 min followed by 5 min at 4 °C. DNA cleavage was monitored on a 20% denaturing acrylamide gel (8 M urea, 1 × TAE buffer) and run at 60 °C for 60 min at 200 V using a BIO-RAD Protean II xi Cell apparatus. DNA oligonucleotide probes were synthetized by Thermo Fisher Scientific. Oligonucleotide sequences used in this study are listed in Supplementary Methods.

### Exonuclease T degradation assay

The exonuclease T degradation assays were performed as described by the manufacturer (New England BioLabs, #M0265). Reactions (20 µl) containing 1 µM of DNA and indicated concentration of Exo T were incubated for 30 min at 25 °C in a reaction buffer (20 mM Tris-Ac pH 7.9, 50 mM KAc, 10 mM MgCl$_2$, 1 mM DTT) followed by 10 min at 95 °C. DNA degradation was monitored on a 20% denaturing acrylamide gel (8 M urea, 1 × TAE buffer) and run at 60 °C for 100 min at 160 V.

### Oligo-Seq

U2OS or HEK-293T cell lysates, depleted for UNG with siRNA, and expressing either A3A or A3B were prepared as described above. Reactions (50 µL) containing 8 µL of normalized amount of U2OS or HEK-293T cell lysates (to not exceed a maximum of 10% deamination efficiency) were incubated for 1 h at 37 °C with a pool of synthetic DNA oligonucleotides containing random bases at the indicated position in a reaction buffer (42 µL: 60 pmol of DNA oligonucleotide), 50 mM Tris-HCl, [pH 7.5], and 10 mM EDTA], followed by 30 min incubation at 90 °C for enzymatic deactivation. Reaction products were then purified and concentrated using Oligo Clean & Concentrator kit (Zymo Research, #D4061) and eluted in 6 µL of 10 mM Tris-HCl (pH 8.0). The purified products were then added to a reaction containing, 20 µM of OLIGO 3' adapter (Supplementary Methods), 100 U of CircLigase (Avantor, #CL4115K), 1X CircLigase reaction buffer, 0.05 mM ATP and 2.5 mM MnCl$_2$ in a final volume of 20 µL and then incubated overnight at 60 °C (Ligation tests with Thermostable 5' APP DNA/RNA Ligase [New England BioLabs, #M0319] and T4 RNA Ligase 2 Truncated KQ [New England BioLabs, #M0373] were performed as indicated by the

manufacturer with 5′ pre-adenylated OLIGO 3′ adapter [New England BioLabs, #M0373]). Next, an equal amount of 2x denaturing loading buffer (1 mM EDTA, 100% formamide, and bromophenol blue) was added to the reaction and separated in a 20% denaturing polyacrylamide gel (8 M urea, 1 × TAE buffer) for 3 h at 250 V using a BIO-RAD Protean II xi Cell apparatus. DNA migration was detected by a 5 min incubation with SYBR Gold Nucleic Acid Gel Stain (Invitrogen #S11494, 1/10,000) and revealed using a Chemidoc MP Immaging System (BIO-RAD). The DNA band corresponding to target oligonucleotide ligated to the adapter was excised and flash frozen in 400 µL of DNA gel extraction buffer (300 mM NaCl$_2$, 10 mM Tris-HCl [pH 8], and 1 mM EDTA). The frozen samples were thawed on an agitator overnight at 25 °C. Then, an equal amount of isopropanol +1.5 µL of GlycoBlue coprecipitant (Invitrogen, #AM9515) were added to the supernatant and incubated for 2 h at −20 °C before precipitation by centrifugation (20,000 $g$ for 30 min at 4 °C). DNA was resuspended in 5 µL nuclease free water (Ambion, #AM9937) and incubated with 1 mM dNTPs, 1 × Phusion buffer, 1 U of high-fidelity Phusion polymerase (New England BioLabs, #M0530, 2000 U/mL), and 1.25 µM of OLIGO-Reverse primer (Supplementary Methods) in a final volume of 25 µL. The reactions were incubated or 5 min at 98 °C, 5 min at 55 °C, and 20 min at 72 °C. Following reverse strand extension, equal amount of nuclease free water was added to each sample and purified using Oligo Clean & Concentrator kit (Zymo Research, #D4061). The samples were eluted in 6 µL of nuclease free water The samples were eluted in 6 µL of nuclease free water and an equal amount of 2x denaturing loading buffer was added for separation on a 20% denaturing polyacrylamide gel as described above. The DNA band corresponding to the circular ssDNA was excised and flash frozen in 400 µL of DNA gel extraction buffer. The DNA was extracted as described above and resuspend in 15 µL of 10 mM Tris-HCl (pH 8). Next the ssDNA was circularized using 100 U of CircLigase (1X CircLigase reaction buffer, 0.05 mM ATP, and 2.5 mM MnCl$_2$) in 20 µL and incubated for 2 h at 60 °C. Finally, a 50 µL PCR reaction was performed using 5 µL of the circularized DNA, 0.2 mM dNTPs, 0.4 µM forward primer and reverse primer (Supplementary Methods), 1X Phusion buffer, and 1 U of Phusion polymerase. The PCR reaction was performed using the following settings: 98 °C for 30 s, following by 12 cycles at 94 °C for 15 s, 55 °C for 5 s, and 65 °C for 10 s. The PCR product was purified using DNA Clean & Concentrator-5 kit (Zymo Research, #11-380) and eluted in 6 µL 10 mM Tris-HCl (pH 8). Then, 14 µL of Novex Hi-Density TBE Sample Buffer was added to the samples and separated in a 10% polyacrylamide non-denaturing gel (Mini-PROTEAN TGX, BIO-RAD, #4561023) using 1X TBE running buffer for 30-35 min at 200 V. DNA migration was detected by incubating the gel with SYBR Gold Nucleic Acid Gel Stain (Invitrogen, #S11494, 1/10,000) in 1X TBE for 5-10 min and reveal using a Chemidoc MP Immaging System (BIO-RAD). The DNA corresponding to the linear double-stranded DNA was excised and precipitated as described above. The DNA samples were resuspended in 5 µL 10 mM Tris-HCl (pH 8). Library size distributions were measured using a BioAnalyzer and quantified via qPCR. Libraries were sequenced on a NovaSeq 6000 platform using PE100 cycle chemistry.

## Oligo-Seq data analysis

For Oligo-seq data analysis, raw sequences were extracted from the fastq sequence files and aligned using stringr package on R statistical software. During alignment, sequences were filtered such that only those with the full oligonucleotide sequence were included in the analysis. After alignment, single-position nucleotide frequency was calculated using Biostrings package. During this analysis, ratio values were obtained by comparing the single-position nucleotide frequency of deaminated sequences versus the total population of each individual sequence (non-deaminated and deaminated sequences), and normalized to a value of 1. The resulting normalized values were used to make a sequence logo plot using ggseqlogo package[93]. Finally, the

frequency of sequences containing specific nucleotide combinations were extracted and analyzed. Ratio values were obtained by comparing the number of deaminated sequences versus the total population (non-deaminated and deaminated sequences) and normalized to a value of 1.

## APOBEC3A and APOBEC3B purification

HEK-293T cells transiently expressing A3A-GFP/Flag or A3B-GFP/Flag were collected and resuspended in lysis buffer [50 mM Tris-HCl (pH 7.5), 150 mM NaCl, 1 mM EDTA, and 0.5% Igepal] containing protease inhibitors (P8340, Sigma) and phosphatase inhibitors [NaF (5 mM) and Na$_3$VO$_4$ (1 mM)], incubated for 5 min on ice, and lysed by sonication. Insoluble material was removed by high-speed centrifugation (20,000 $g$ at 4 °C). RNase A (0.2 mg/ml) was added and incubated for 30 min at 4 °C and insoluble material was removed by high-speed centrifugation (20,000 $g$ at 4 °C). Then, 100 µl of M2 anti-Flag affinity gel (Millipore Sigma, #A2220) was added to the soluble extract for 2 h 30 min at 4 °C. The beads were then washed three times with washing buffer (50 mM Tris-HCl, pH 7.5, 350 mM NaCl, 2 mM EDTA, and 0.5% Igepal) and one time with high salt buffer (50 mM Tris-HCl, pH 7.5, 500 mM NaCl, 2 mM EDTA, and 0.5% Igepal) followed by two additional washes with elution buffer [25 mM HEPES (pH 7.9), 10% glycerol, 150 mM NaCl, 1 mM EDTA, 1 mM MgCl$_2$, 1 mM ZnCl$_2$]. Finally, A3A or A3B were eluted in 200 µl of elution buffer containing 3 × Flag peptide (Millipore Sigma, #SAE0194, 500 µg/ml) for 3 h at 4 °C. A3A and A3B purification were validated by Western blotting. Purified proteins were aliquoted and stored at −80 °C.

## Bioinformatic analyses of mouse and human tumors

The analysis of mutated hairpins was performed as described in refs. 15,16. The mouse genome (build mm9) and human genome (build hg19) were scanned for potential hairpin-forming sequences using the survey_hairpins function of the ApoHP tool [http://github.com/alangenb/ApoHP], which implements a version of the algorithm described in previous work[15,16]. Briefly, the genome was scanned for sequences of the form S-L-S′, where the sequences S and S′ are reverse-complementary with a sequence L (ranging from 3 to 11 nucleotides) intervening between them. Sequences such as these have the potential to form stem-loop, or "hairpin" structures in DNA that is transiently single-stranded. For each position p in the genome, flanking sequences S and S′ were sought such that position p would be in the intervening loop sequence L. Stem strength was defined as the number of A:T basepairs plus 3× the number of G:C basepairs, an approximation of empirically measured nearest-neighbor stacking energies[94]. In cases where multiple alternative pairings were possible, the stem with the strongest pairing was chosen, using shortest loop size as a tie-breaker. The output of this procedure was to assign to each genomic position a set of parameters describing its hairpin characteristics: stem strength, loop length (in nucleotides), and position of the mutation-site cytosine within the loop (ranging from 1 to loop length). This allows genomic positions to be categorized into equivalence classes for investigating the influence of hairpin characteristics on relative mutation frequency. Additional information on the bioinformatic analyses of mouse and human tumors is described in the Supplementary Methods.

## Statistics and reproducibility

All western blots and DNA gels showed in Figs. 1e, 3a, b, 4d, 5a, b, 5d–f and Supplementary Figs. 1C–H, 2A–C, 3A, B, 3D, 4B–I, 5A, 6C–E, 7C, and 8A, B were repeated at least three times and representative images are shown in this paper.

## Reporting summary

Further information on research design is available in the Nature Portfolio Reporting Summary linked to this article.

## Data availability

Sequencing reads from mouse tumors are available at the NCBI's Sequence Read Archive (SRA) under BioProject ID: PRJNA927047 and PRJNA655491. Sequencing data generated from Oligo-seq experiments in this study are available at the NCBI's SRA under BioProject ID: PRJNA1010353. Source data are provided with this paper.

## Code availability

Source code and executable software tool ApoHP are available at http://github.com/alangenb/ApoHP[16].

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

## Acknowledgements

We thank, Casey Johnson and Melanie Oakes for their technical assistance and Dr. Ashok Bhagwat (Wayne State University) for sharing preliminary results and helpful discussion. A.S. is supported by a National Institutes of Health Research Supplements to Promote Diversity in Health-Related Research (R37-CA252081-S1). Salary support for P.O. was provided by a California Institute for Regenerative Medicine (CIRM) stem cell biology training grant (TG2-01152) and an EMBO Postdoctoral fellowship (ALTF 213-2022). L.M. is supported by a Center for Virus Research Graduate Fellowship funded by the UCI Division of Graduate Studies. S.O. is a Dr. Lorna Calin Scholar and was supported by the Faculty Mentor Program from the University of California, Irvine. C.D. is supported by a CPRIT Research Training Award (RP 170345). This work was supported by NCI R37-CA252081 (R.B.), NIAMS P30AR075047 seed grant (R.B.), NIAID R01 AI150524 (X.S.C.), NCI P01 CA234228 (R.S.H.), and a Recruitment of Established Investigators Award from the Cancer Prevention and Research Institute of Texas (CPRIT RR220053 to R.S.H.). R.S.H. is an investigator of the Howard Hughes Medical Institute and the Ewing Halsell President's Council Distinguished Chair. This work was also made possible, in part, through access to the UCI Genomics Research and Technology (GRT) Hub parts of which are supported by NIH grants to the Chao Family Comprehensive Cancer Center (P30CA-062203) as well as to the GRT Hub for instrumentation (1S10OD010794-01 and 1S10OD021718-01).

## Author contributions

A.S., P.O., and R.B. designed the experiments. A.S., P.O., L.M., S.O., E.B., A.B., and R.B. performed the experiments. P.O. developed the Oligo-seq method, and A.S. performed Oligo-seq experiments shown in this study. C.D., N.A.T., and R.S.H. provided mouse whole genome sequencing data. R.S. and M.S.L performed the bioinformatic analyses on mouse and human whole genome sequencing data. K.K. and X.C. provided A3B and A3A chimera constructs. R.B. conceived the study and oversaw the project. R.B. wrote the paper, and all authors contributed to manuscript revisions.

## Competing interests

R.B. has served as a compensated consultant for Pfizer and Health Advances. The remaining authors declare no competing interests.

## Additional information

[1]Department of Biological Chemistry, School of Medicine, University of California Irvine, Irvine, CA, USA. [2]Center for Epigenetics and Metabolism, Chao Family Comprehensive Cancer Center, University of California Irvine, Irvine, CA, USA. [3]Massachusetts General Hospital Cancer Center, Harvard Medical School, Boston, MA, USA. [4]Broad Institute of Harvard and MIT, Cambridge, MA, USA. [5]Molecular and Computational Biology, Departments of Biological Sciences and Chemistry, University of Southern California, Los Angeles, CA, USA. [6]Department of Biochemistry and Structural Biology, University of Texas Health San Antonio, San Antonio, TX, USA. [7]Institute for Health Informatics, University of Minnesota, Minneapolis, MN, USA. [8]Masonic Cancer Center, University of Minnesota, Minneapolis, MN, USA. [9]Howard Hughes Medical Institute, University of Texas Health San Antonio, San Antonio, TX, USA. [10]Department of Pathology, Massachusetts General Hospital, Harvard Medical School, Boston, MA, USA. [11]Department of Pharmaceutical Sciences, School of Pharmacy & Pharmaceutical Sciences, University of California Irvine, Irvine, CA, USA. ✉e-mail: rbuisson@uci.edu

