## [Peer Review File · Nature Communications]

REVIEWER COMMENTS

Reviewer #1 (Remarks to the Author):

In this manuscript, Sanchez and colleagues examine the specific differences between APOBEC3A (A3A) and A3B DNA substrate preferences. Focusing on DNA stem-loops, the authors determine the bases context around cytosine bases mutated by the two APOBEC3 enzymes. The authors develop a new technique, Oligo-seq, to facilitate higher throughput identification of A3A and A3B substrate preferences. All Oligo-seq findings are validated by in vitro deamination experiments. While the authors have previously established that A3A mutates cytosine bases in the 3'-most position of 3-nucleotide stem-loops, they now show that A3B preferentially mutates similarly located cytosine bases in larger (4-5 nucleotide) loops using in vitro assays. Further, they determine this difference in substrate specificity is due to the amino acid sequence of loop one in A3A and A3B. Analysis of mutations occurring after A3B expression in mice, as well as analysis from human tumors exhibiting mutations in the previously defined A3B trinucleotide context, shows that A3B acts on cytosine bases in stem loops. This paper is one of the first reports characterizing A3B mutations in DNA stem loops

Overall this is a well written manuscript which entails rigorous, extensive biochemical experiments. Oligo-seq is an innovative technique that has the potential to enable additional experiments. Beyond the biochemistry, analysis of distinctions between A3A and A3B are somewhat shallow. In mouse and human tumor sequences, the differences are less clear. Further, the impact of this subtle distinction between A3A and A3B preferring 3 or 4 nt loops respectively is unclear.

Specific comments:

-Most notably, Fig 6 seems to show only modest overlap in stem loop mutations between mouse/human genomes and in vitro data. For example, the mouse tumors with A3B appear to have similar frequency of mutations in 3nt, 4nt, and 5nt stem loops (maybe even 6nt stem loops) whereas A3B+ human tumors have increased mutations only in 3-4nt loops. Further, the relative mutation frequency (y-axis) is quite small for A3B tumors, so it is unclear what conclusions can be drawn from very few mutations. Statistical analysis for this figure would be helpful to support the authors' interpretation of a "strong prevalence" for A3B at hairpins in human tumors and a "striking similarity" between mouse and human tumors (lines 415, 418). My interpretation of the presented data is that A3A and A3B have similar loop size preferences in vivo, which may only be distinguishable from one another by the RTC or YTC context (a preference already characterized outside of the context of stem-loops).

-Please define what hairpin mutation character means in Fig 6F. It is unclear how this is represented as a spectrum.

-Within the Methods- Cell Culture section, the authors note that "U2OS-derived cell lines expressing APOBEC3B were generated by infecting U2OS cells with lentivirus expressing APOBEC3B under a doxycycline-inducible promoter..." The manuscript only discusses endogenous expression of A3B in U2OS cells. It would be helpful if the authors could clarify which system is being used.

Additional minor suggestions:

-Different amounts of A3A and A3B-containing lysate are used throughout the manuscript (often much more lysate from A3B-expressing cells than from A3A). Are the levels of A3A and A3B comparable at these levels?

-Line 65: "...has established them as significant drivers of cancer." To my knowledge A3A and A3B have been shown to accelerate tumor formation in mice, though data for APOBECs driving human cancers have not yet been defined. Perhaps soften this statement so as not to overinterpret the literature.

-It would be helpful to define the term mesoscale for unfamiliar readers.

- Line 135- the word "preformed" is used when I believe "preferred" is meant.

- The term "single-stranded (ss) DNA" is used throughout the manuscript when "linear" is meant.

- Lines 534-535 "Ultimately, CpG methylation has resulted in a decrease of available TpCpG motifs for A3A- and A3B-induced deamination" The discussion of relative infrequency of CpG sites (lines 523-542) as a limitation to defining G as a +1 preference for APOBEC substrate can be solved by defining the denominator. If the number of CpG sites (or TCpG sites) in a genome is known, then the relative frequency of mutation of those sites can be determined.

-Supp Fig 3C: the hairpin loop is mislabeled, the bases in blue should be CA rather than CG

Reviewer #2 (Remarks to the Author):

#NCOMMS-23-34689

The manuscript by Sanchez and Colleagues analyzes the features that affect the targeting of APOBEC3A and APOBEC3B, two deaminases involved in the onset of cancer mutations. To this aim, the Authors use a series of biochemical, molecular and bioinformatic approaches to characterize APOBEC3A and APOBEC3B biochemical properties. Among them, the Authors develop Oligo-seq, a new in vitro sequencing approach to determine the sequence contexts that facilitate the activity of these two deaminases. Finally, the Authors show that the targeting preferences of APOBEC3A and APOBEC3B can discriminate the weight of each deaminase in the different cancers.

Overall, the manuscript is very readable, and the work is flawless, with a logical progression of experiments that supports each claim and foretells any question/doubt I could have. Beyond the experimental part, the introduction is remarkable for its comprehensive and insightful background (with all relevant references!)

The relevance and importance of manuscripts lies in the light it sheds on the ongoing debate on the role of these enzymes in cancer.

Major comments:

1. while I am not worried about the biological relevance of the findings (the Authors support their claims through multiple converging approaches), there is an overall lack of statistics, especially in the bioinformatic analyses on the cancer datasets.

Minor comments:

1. There is an apparent discrepancy between the Results and the Methods on the use of U2OS cells. In most of the text, it is always specified the wild-type U2OS cells were used, as they already express A3B (but in Suppl. Fig 4, where FLAG-tagged A3B expressing U2OS are mentioned). In the Methods, it seems that the U2OS cells were engineered to overexpress A3B by viral transduction.

2. While it does not affect the results or their interpretation, it would have been logical to use A3A and A3B produced in the same way (i.e. overexpression in HEK293T cells).

Overall, the Authors do a good job explaining the rationale behind the experimental setup but, in a few spots, more explanation would be helpful:

4. What is the reason for choosing a 5-base-pair stem and its importance for the Oligo-seq sequencing approach? Does the sequence of the stem affect the assay?

5. Why was the NUP93 gene chosen for the synthetic DNA substrates?

6. whole-cell extracts were titrated to restrict reactions to a maximum of 10% completion. It would be helpful to explain the reasoning behind this choice and why it is crucial for the experiment.

7. I wonder whether it might be worth discussing the BioRxiv manuscript by Vega et al. (10.1101/2022.06.01.494353) in light of your findings (also with regard to the introduction, page 4, lines 77 and 84).

8. in the description of Oligo-seq analysis - "single-position nucleotide frequency of deaminated sequences versus the total population (non-deaminated and deaminated sequences), and normalized to a value of 1" (page 35, line 894) - I assume the meaning for "total population" refers to the total population for a single sequence (edited or not).

9. Have the Authors tested the complexity of the oligo library used, or did they rely on the sole ratio between edited and non-edited reads for a given oligo?

10. It might be helpful to explain the logograms generated with ggseqlogo (e.g. what it means enrichment/depletion)

11. The Authors mention extensive testing of DNA ligases to set up the Oligo-seq assay. It might be worth reporting a discussion of the tests performed in the supplementary information.

12. I am not sure the discussion on the rarity of CpG dinucleotides makes sense: even if rarer, an enrichment for TpCpG targeting should become apparent after normalization. This does not seem the case. I appreciate the potential overlap with the signature from spontaneous deamination, but there might be more to it (e.g. targeting being affected by methylation)

13. Figure 4: Considering the Authors already use colors for the different contexts (Fig 4D), matching the colors in Fig 4E would improve its readability.

14. Figure 5D-F: while loop1 is indeed critical for substrate selectivity, swapping it between A3A and A3B does not swap completely its specificity (e.g. A3B[a3Aloop] is not identical to wild type A3A). This suggests that there might be other elements playing a role in substrate specificity.

Reviewer #3 (Remarks to the Author):

In this manuscript, the authors investigated substrate preference for APOBEC3B (A3B) for the stated purpose of further distinguishing mutations caused by A3B versus mutations caused by APOBEC3A (A3A). They additionally define some of the structural differences between the two enzymes that explains their distinct (expanded) mutation signatures on ssDNA 2° structures. In addition to determining biochemical substrate selectivity in vitro deaminase activity assays, the authors also analyzed mutations in mouse and human tumors to determine whether the innate substrate preference of these two enzymes is influenced by cellular factors.

The authors first investigated A3B vs A3A preference for deaminating cytidines within hairpin loops of varying loop size vs linear ssDNA, determining in conflict with a previous report from the same group (<https://www.science.org/doi/10.1126/science.aaw2872>) that A3B also prefers hairpins over linear ssDNA. The authors observed higher A3B activity on larger loops of 4 and 5nt, whereas A3A prefers smaller, 3nt loops (consistent with previous reports). By changing the positioning of cytidine within these larger loops, the authors were able to further conclude that A3B prefers 3' cytidine positioning. A3A was also found to deaminate larger loops with less efficiency but displays a marked preference for cytidine at position 4 within 5 nt loops in vitro. Results obtained using whole cell lysate was recapitulated using purified A3A and A3B. The authors also analyzed A3B- versus A3A-induced mutations in mouse tumors and observed similar respective substrate preferences as in their previous experiments, however A3B substrate preference appears to be dampened in this model, both in terms of preferring larger loops and requiring 3' C positioning within these loops. The authors concluded that this is likely due to the enhanced stability of small loop-forming hairpins in cells and cited cellular factors as a likely explanation for this observation. The authors also analyzed APOBEC mutations identified from whole genome sequencing of human tumors and further analyzed sequence context of mutations in tumors previously identified to be A3B vs A3A-preferred (using previously established preferences for R or Y respectively at the -2 position), and observed similar preferences for small hairpin loops with 3' C positioning for both enzymes as those observed in mouse tumors.

The authors investigated how differing structures between A3A and the C-terminal domain of A3B influence their different substrate preferences by replacing loop 1 (located in close proximity to both enzymes' active sites) of A3B with the more truncated/restrictive loop 1 of A3A and observed a stronger preference for smaller hairpin loops as a result. Conversely replacement of A3A loop 1 with that of A3B

significantly reduced activity on smaller hairpin loops but increased activity on larger loops resembling A3B's preferred substrate. The authors concluded that loop 1 plays a critical role in influencing the preference of these two enzymes for different substrates.

The authors also expanded on previous APOBEC literature by extending their analysis of mutations caused by A3B or A3A to the 5' end of the canonical APOBEC mutation signature to identify distinct signatures of each enzyme. Notably the authors used a deconvolution method, testing A3B and A3A activity on hairpin-forming oligonucleotides, systematically altering bases at the -2, -3, and -4 positions (in combination). Using this method they were able to identify a clear preference of A3B and A3A for loops where cytidine is preceded by a YYRT and YYT motif respectively (where Y is a pyrimidine and R is a purine), with a strong preference for C at the -4 position for both enzymes.

This manuscript provides a moderate advance in the field of APOBEC cancer biology by further defining A3B and A3A substrate specificity. The methodologies used were sound and most experiments were adequately rigorous, with reasoned conclusions being made, however statistical comparisons are generally lacking throughout the manuscript.

Specific critiques:

1. Little to no rationale is given to explain why the previous report (<https://www.science.org/doi/10.1126/science.aaw2872>) indicated that A3B preferred linear sequences over hairpin.
2. Please include a gel (i.e. Coomassie, silver stain) that indicates purity of the purified A3A and A3B enzymes as contaminating proteins may influence enzyme substrate preference.
3. Please add a better description of what is being compared for the "relative mutation frequency" in the y-axis on figures 6 a,b,d, and e to the figure legend.
4. A figure indicating how many A3A and A3B-induced mutations were observed in mouse and human tumors in hairpins versus non-hairpin forming sequences would be very helpful in evaluating the importance of hairpin mutagenesis in cancer. The data in Supplemental figure 8 c and d seem to indicate that hairpin mutations caused by A3B are very few.
5. If A3B is not making many mutations in mouse tumors, could the resulting lack of signal be one reason for the less prominent A3B hairpin signature in mice and human tumors, instead of other cellular factors (as suggested by the authors)?
6. On lines 459-462, the authors claim they can use the mesoscale preferences of A3A and A3B to identify which enzymes are active in patient tumors. An analysis comparing the utilization of the mesoscale characteristics versus the previously used YTCA versus RTCA sequence specificity should be used to determine if the mesoscale preferences have added value in this differentiation.

7. Do A3B targeted hairpins also cause recurrent mutations in human or mouse tumors like A3A targeted hairpins do? Are any A3B targeted hairpin mutations potential drivers or are they primarily passenger mutations too?

8. The utility of the detailed analysis of A3A and A3B substrate specificity appears limited in regards to translational cancer biology. However, the authors indicate that this information may be very useful for designing better base editors at the end of the discussion. This idea appeared to not be well integrated into the rest of the manuscript, but could provide a larger impact for the work.

REVIEWER COMMENTS

Reviewer #1 (Remarks to the Author):

In this manuscript, Sanchez and colleagues examine the specific differences between APOBEC3A (A3A) and A3B DNA substrate preferences. Focusing on DNA stem-loops, the authors determine the bases context around cytosine bases mutated by the two APOBEC3 enzymes. The authors develop a new technique, Oligo-seq, to facilitate higher throughput identification of A3A and A3B substrate preferences. All Oligo-seq findings are validated by in vitro deamination experiments. While the authors have previously established that A3A mutates cytosine bases in the 3'-most position of 3-nucleotide stem-loops, they now show that A3B preferentially mutates similarly located cytosine bases in larger (4-5 nucleotide) loops using in vitro assays. Further, they determine this difference in substrate specificity is due to the amino acid sequence of loop one in A3A and A3B. Analysis of mutations occurring after A3B expression in mice, as well as analysis from human tumors exhibiting mutations in the previously defined A3B trinucleotide context, shows that A3B acts on cytosine bases in stem loops. This paper is one of the first reports characterizing A3B mutations in DNA stem loops. Overall this is a well written manuscript which entails rigorous, extensive biochemical experiments. Oligo-seq is an innovative technique that has the potential to enable additional experiments. Beyond the biochemistry, analysis of distinctions between A3A and A3B are somewhat shallow. In mouse and human tumor sequences, the differences are less clear. Further, the impact of this subtle distinction between A3A and A3B preferring 3 or 4 nt loops respectively is unclear.

We thank the reviewer for his/her appreciation of the significance and quality of our work. We have now addressed all reviewer's comments.

Specific comments:

-Most notably, Fig 6 seems to show only modest overlap in stem loop mutations between mouse/human genomes and in vitro data. For example, the mouse tumors with A3B appear to have similar frequency of mutations in 3nt, 4nt, and 5nt stem loops (maybe even 6nt stem loops) whereas A3B+ human tumors have increased mutations only in 3-4nt loops. Further, the relative mutation frequency (y-axis) is quite small for A3B tumors, so it is unclear what conclusions can be drawn from very few mutations. Statistical analysis for this figure would be helpful to support the authors' interpretation of a "strong prevalence" for A3B at hairpins in human tumors and a "striking similarity" between mouse and human tumors (lines 415, 418). My interpretation of the presented data is that A3A and A3B have similar loop size preferences in vivo, which may only be distinguishable from one another by the RTC or YTC context (a preference already characterized outside of the context of stem-loops).

We agree with the reviewer that both A3A and A3B showed a preference for 3-nt and 4-nt loop hairpins in mouse and human tumor samples. However, the ratio of mutation

levels between both loop lengths is very different when mutated by A3A or A3B. A3A-induced mutations in hairpins elicited a strong preference for 3-nt loops versus 4-nt, while A3B mutates more hairpins with a 4-nt loop than a 3-nt loop. Based on this ratio difference, we established the “hairpin mutation character” that allowed us to identify which patients’ tumors accumulate APOBEC mutational signatures driven by A3A or A3B. To further support the results from the mouse and human tumors, an accompanying paper from Dr. Ashok Bhagwat’s Lab (Butt et al. BioRxiv. <https://doi.org/10.1101/2023.08.01.551518>) used a bacteria system to monitor A3B DNA sequences and substrate preferences. Similar to our data, Bhagwat’s lab found that mutations caused by A3B in hairpins are also strongly enriched in 4-nt loops with similar sequence context, while A3A preferentially mutates 3-nt loop hairpins. Together, our studies strongly support the difference in stem-loop mutation patterns caused by A3A or A3B in genomic DNA.

We would like to point out that identifying tumors with APOBEC mutations driven by A3A or A3B was not possible based solely on the RTC versus YTC context. Indeed, the graph shown in Figure 6C and 6F, illustrates how tumor samples present in the center of the graph accumulate a significant number of APOBEC-induced mutations without eliciting any clear preference for either RTC or YTC context. Thus, it was important for us to determine by another means the mutations that are caused by A3A or A3B.

Following the reviewer's comment, we have now added statistical analysis to the panels of **Figures 6A, B, D, and E**. We calculated the p-values by performing a binomial test for each group and we adjusted these p-values for multiple comparisons using the Benjamini & Hochberg method. Note that we were not able to detect a significant enrichment of mutations in 3-loop hairpins for A3B tumors using this method, further suggesting that A3A and A3B mutate different types of hairpins in genome tumors. In addition, we performed a Pearson correlation analysis between the normalized mutation rates of mouse samples versus human tumors to further support our claim that there is a high similarity of mutational patterns between mouse and human tumors (no significant correlations were found when comparing A3A versus A3B driven tumors). We have now added these values to the manuscript.

A3B mouse tumors versus A3B human tumors: p-value 0.000924, correlation 0.7127
A3A mouse tumors versus A3A human tumors: p-value 1.482e-12, correlation 0.9795

-Please define what hairpin mutation character means in Fig 6F. It is unclear how this is represented as a spectrum.

We would like to apologize for the lack of detail on how we define hairpin mutation character. We have now added more information in the manuscript on how we defined the hairpin mutation character in **Figure 6F**. “*To determine A3A or A3B mutation characters, we calculated the ratio between the levels of 4-nt hairpin loop and 3-nt hairpin loop mutated by A3B and A3A respectively, multiplied by the ratio of mutated ATC versus TTC sites (motifs targeted by A3B and A3A respectively) present in the 4-nt hairpin loops with TpC motifs positioned at the 3' end.*”

-Within the Methods- Cell Culture section, the authors note that “U2OS-derived cell lines expressing APOBEC3B were generated by infecting U2OS cells with lentivirus expressing APOBEC3B under a doxycycline-inducible promoter...” The manuscript only discusses endogenous expression of A3B in U2OS cells. It would be helpful if the authors could clarify which system is being used.

We want to apologize for the mistake in the method section of our manuscript (due to a copy-paste of our method section from a previous manuscript) and thank the reviewer for catching it. In this study, we used U2OS (wild type) to generate cell extract that expressed endogenous levels of A3B. This U2OS cell extract was used throughout this study. Only in a few panels where we compared A3B wild-type to A3B mutants (e.g., Fig. 5D-F), we expressed A3B wild-type and mutants in HEK293T cells as described in our method section “*Cell Lysate Preparation*” and the corresponding figure legends. We have now removed the sentence referring to the U2OS cell line expressing APOBEC3B under a doxycycline-inducible promoter from the manuscript.

Additional minor suggestions:

-Different amounts of A3A and A3B-containing lysate are used throughout the manuscript (often much more lysate from A3B-expressing cells than from A3A). Are the levels of A3A and A3B comparable at these levels?

This is a good point. A3A and A3B are known to have very different activity levels. A3A has been shown to have a much higher deaminase activity than A3B (PMID: 23104058 , PMID: 29234087). In this study, we aimed to focus on characterizing A3A and A3B selectivity activity (enzyme's capacity to discern and differentiate between various substrates) rather than catalytic activity (efficiency of an enzyme in catalyzing a specific biochemical reaction). Thus, we had to use a higher amount of cell extract expressing A3B to compensate for its low deaminase activity and to reach a similar catalytic activity level as A3A.

-Line 65: “...has established them as significant drivers of cancer.” To my knowledge A3A and A3B have been shown to accelerate tumor formation in mice, though data for APOBECs driving human cancers have not yet been defined. Perhaps soften this statement so as not to overinterpret the literature.

We agree with the reviewer. We have now revised our statement: “*The ability of A3A and A3B to rewrite genomic information has established them as significant drivers of diversity and heterogeneity within tumor genomes*”

-It would be helpful to define the term mesoscale for unfamiliar readers.

This is a good point! We have now added a sentence in the manuscript to define the term mesoscale. *“To interrogate how A3A and A3B activities are impacted by mesoscale genomic features – characterized by DNA sequences ranging from 3- to 30-base pair length with the capacity to adopt various structural configurations, ...”*

- Line 135- the word “performed” is used when I believe “preferred” is meant.

We used the term “performed” to point out that when the DNA already forms a U-shape structure by folding into a hairpin it will favor the interaction with the active site of A3B. For clarity, we have now changed our statement: *“This suggests that U-shaped structures already formed in DNA stem-loops are likely more favorable than linear DNA for deamination by A3B.”*

- The term “single-stranded (ss) DNA” is used throughout the manuscript when “linear” is meant.

We agree with the reviewer that linear is a better term since hairpins are also formed by ssDNA. We have now made this change throughout the manuscript.

- Lines 534-535 “Ultimately, CpG methylation has resulted in a decrease of available TpCpG motifs for A3A- and A3B-induced deamination” The discussion of relative infrequency of CpG sites (lines 523-542) as a limitation to defining G as a +1 preference for APOBEC substrate can be solved by defining the denominator. If the number of CpG sites (or TCpG sites) in a genome is known, then the relative frequency of mutation of those sites can be determined.

Based on the comments from reviewers 1 and 2, we have now removed this speculative paragraph from the discussion of the manuscript. We agree with reviewers that additional analyses will be required to make this conclusion and the current discussion was very hypothetical. We hope the reviewer will agree that how CpG methylation impacts the decrease of available TpCpG motifs is a question beyond the scope of the current study.

-Supp Fig 3C: the hairpin loop is mislabeled, the bases in blue should be CA rather than CG

Thanks for this catch! We have now corrected this typo.

Reviewer #2 (Remarks to the Author):

The manuscript by Sanchez and Colleagues analyzes the features that affect the targeting of APOBEC3A and APOBEC3B, two deaminases involved in the onset of cancer mutations. To this aim, the Authors use a series of biochemical, molecular and bioinformatic approaches to characterize APOBEC3A and APOBEC3B biochemical properties. Among them, the Authors develop Oligo-seq, a new in vitro sequencing approach to determine the sequence contexts that facilitate the activity of these two deaminases. Finally, the Authors show that the targeting preferences of APOBEC3A and APOBEC3B can discriminate the weight of each deaminase in the different cancers.

Overall, the manuscript is very readable, and the work is flawless, with a logical progression of experiments that supports each claim and foretells any question/doubt I could have. Beyond the experimental part, the introduction is remarkable for its comprehensive and insightful background (with all relevant references!) The relevance and importance of manuscripts lies in the light it sheds on the ongoing debate on the role of these enzymes in cancer.

We would like to deeply thank the reviewer for his/her appreciation of our introduction as well as the significance and quality of our work. We have now addressed all the reviewer's comments.

Major comments:

1. while I am not worried about the biological relevance of the findings (the Authors support their claims through multiple converging approaches), there is an overall lack of statistics, especially in the bioinformatic analyses on the cancer datasets.

We would like to apologize for the lack of statistics in our first submission. We have now added additional statistic analysis throughout the manuscript including in the bioinformatics analyses on the cancer datasets (**Figure 6**).

Minor comments:

1. There is an apparent discrepancy between the Results and the Methods on the use of U2OS cells. In most of the text, it is always specified the wild-type U2OS cells were used, as they already express A3B (but in Suppl. Fig 4, where FLAG-tagged A3B expressing U2OS are mentioned). In the Methods, it seems that the U2OS cells were engineered to overexpress A3B by viral transduction.

We want to apologize for the mistake in the method section of our manuscript (due to a copy-paste of our method section from a previous manuscript) and thank the reviewer for catching it. In this study, we used U2OS (wild-type) to generate cell extract that

expressed endogenous levels of A3B. This U2OS cell extract was used throughout this study. Only in a few panels where we compared A3B to A3B mutants (e.g., Fig. 5D-F), we expressed a FLAG-tagged A3B WT and mutants in HEK293T cells as described in our method section “*Cell Lysate Preparation*” and the corresponding figure legends. In addition, we purified A3B from HEK293T overexpressing a FLAG-tagged A3B. The Flag tag was used to pulldown A3B with a Flag antibody during the purification method. We have now removed the sentence referring to the U2OS cell line expressing APOBEC3B under a doxycycline-inducible promoter and updated the manuscript accordingly.

2. While it does not not affect the results or their interpretation, it would have been logical to use A3A and A3B produced in the same way (i.e. overexpression in HEK293T cells).

Sorry for the lack of clarity in our manuscript concerning the different cell extracts we tested. We did confirm that A3B overexpressed in HEK293T cells produced the same results as the cell extract expressing endogenous levels of A3B. For example, the deamination results showed in **Figure 5B** were performed with cell extract expressing endogenous A3B, whereas the deamination results showed in **Figure 5D** were performed with HEK293T cell extract overexpressing A3B or A3B mutant. Both types of cell extract showed the exact same result. However, we thought focusing on cell extract expressing endogenous levels of A3B was more relevant for this study when we were not comparing A3B wild-type to A3B mutants.

Overall, the Authors do a good job explaining the rationale behind the experimental setup but, in a few spots, more explanation would be helpful:

4. What is the reason for choosing a 5-base-pair stem and its importance for the Oligo-seq sequencing approach? Does the sequence of the stem affect the assay?

This is a good point! We selected a 5-base-pair stem hairpin to avoid the formation of a hairpin with a stem that might be strong enough to block the DNA polymerase used to perform the library preparation that converted the ssDNA to a dsDNA (Figure 2, step 3). During the optimization of our method, we also performed an oligo-seq with a shorter base-pair stem (3-bp). We obtained a very similar ranking of the sequences with only slight differences that may be attributed to the presence of a less stable stem in the oligonucleotides. We have now added this sentence in the manuscript: “*For this assay, we opted for a 5-bp stem size hairpin, which was sufficient to block ExoT activity (Supplementary Figure 2B), to best facilitate the DNA synthesis carried out by the DNA polymerase as illustrated in step 3 of the library generation process (Figure 2A) and potentially preventing the DNA stem-loop from obstructing the polymerase.*”

5. Why was the NUP93 gene chosen for the synthetic DNA substrates?

In our original study, when we first reported that A3A targets DNA stem-loop structures, we tested our A3A deaminase in vitro assay using several hotspot mutations identified

in patient tumors (Buisson et al. Science, 2019). We selected the mutated hairpin found in the NUP93 gene as a model hairpin-forming sequence to study how A3A recognized DNA stem-loop structures. Since then, we and other research groups have used the hairpin present in the NUP93 gene as the backbone of the synthetic DNA substrate to study A3A activity. We have now added an additional explanation to our manuscript: *“This hairpin-forming sequence was previously identified to be mutated in several patient tumor samples with high levels of APOBEC mutations.”*

6. whole-cell extracts were titrated to restrict reactions to a maximum of 10% completion. It would be helpful to explain the reasoning behind this choice and why it is crucial for the experiment.

This is another important point! We have now added an additional explanation for why we titrated the WCE to restrict reactions to a maximum of 10% completion: *“It is crucial to conduct the reaction under these limiting conditions to favor deamination on the optimal substrates specifically.”*

7. I wonder whether it might be worth discussing the BioRxiv manuscript by Vega et al. (10.1101/2022.06.01.494353) in light of your findings (also with regard to the introduction, page 4, lines 77 and 84).

We have added this reference to our manuscript. Thanks for suggesting it.

8. in the description of Oligo-seq analysis - “single-position nucleotide frequency of deaminated sequences versus the total population (non-deaminated and deaminated sequences), and normalized to a value of 1” (page 35, line 894) - I assume the meaning for “total population” refers to the total population for a single sequence (edited or not).

Yes, that is correct. We calculated the ratio for each individual sequences and the total population refers to an individual sequence that is edited or not. We have now modified our method section to improve the clarity of this sentence.

9. Have the Authors tested the complexity of the oligo library used, or did they rely on the sole ratio between edited and non-edited reads for a given oligo?

For the analysis of the Oligo-seq assay, because the reaction contains a pool of oligonucleotides where only the base at the indicated “N” position(s) and the edited TpC site are variable, our oligo library complexity is low. We obtained hundreds of thousands of reads for each sequence in our total population showing a complete coverage and distribution of each different sequence’s possibility and limiting the consequence of the presence of duplicates in our sequencing results.

10. It might be helpful to explain the logograms generated with ggseqlogo (e.g. what it means enrichment/depletion)

We added enrichment/depletion in the Y-axis legend to help the readers who are not familiar with logograms plot to better understand which sequences were found enriched in our oligo-seq results. We have now included information in the figure legend about the meaning of enrichment/depletion: “A sequence logogram showing the fold enrichment and depletion for each of the four DNA bases at the indicated position after deamination of the cytosine by A3B or A3A”

11. The Authors mention extensive testing of DNA ligases to set up the Oligo-seq assay. It might be worth reporting a discussion of the tests performed in the supplementary information.

We have now included a panel in **Supplementary Figure 2A** to illustrate the results of our test using various ligases. From the 3 ligases we tested and known to ligate DNA, only the CircLigase showed efficient ligation of the two single-strand DNAs.

12. I am not sure the discussion on the rarity of CpG dinucleotides makes sense: even if rarer, an enrichment for TpCpG targeting should become apparent after normalization. This does not seem the case. I appreciate the potential overlap with the signature from spontaneous deamination, but there might be more to it (e.g. targeting being affected by methylation)

Based on the comments from reviewers 1 and 2, we have now removed this speculative paragraph from the discussion of the manuscript. We agree with reviewers that additional analysis will be required to make this conclusion and the current discussion was very hypothetical. We hope the reviewer will agree that how CpG methylation impacts the decrease of available TpCpG motifs is a question beyond the scope of the current study.

13. Figure 4: Considering the Authors already use colors for the different contexts (Fig 4D), matching the colors in Fig 4E would improve its readability.

We have now changed the color code of the **Figure 4D** as suggested by the reviewer.

14. Figure 5D-F: while loop1 is indeed critical for substrate selectivity, swapping it between A3A and A3B does not swap completely its specificity (e.g. A3B[a3Aloop] is not identical to wild type A3A). This suggests that there might be other elements playing a role in substrate specificity.

Following the reviewer's comment, we performed additional analyses on A3B to determine which other elements are playing a role on its substrate specificity. Another key distinction between A3A and A3B lies in the presence of an NTD on A3B. We found that the deletion of the NTD ($A3B^{\Delta NTD}$) did not affect A3B's preference for 5-nt versus 3-nt hairpin loops (**new Figure 5F**), demonstrating that the NTD domain has no impact on A3B substrate selectivity. However, the fusion of A3B-NTD to the N-terminal of A3A to generate a chimera protein ($A3B^{NTD-A3A}$) mimicked the activity observed for $A3B^{A3A-loop1}$, whereas the deletion of the NTD from $A3B^{A3A-loop1}$ ($A3B^{\Delta NTD-A3A-loop1}$) mirrored the activity selectivity of A3A wild-type (**new Figure 5F and Supplementary Figure 8B**). These new results suggest that both the loop 1 and the NTD domain need to be modified on A3B to recapitulated the substrate specificity of A3A.

Reviewer #3 (Remarks to the Author):

In this manuscript, the authors investigated substrate preference for APOBEC3B (A3B) for the stated purpose of further distinguishing mutations caused by A3B versus mutations caused by APOBEC3A (A3A). They additionally define some of the structural differences between the two enzymes that explains their distinct (expanded) mutation signatures on ssDNA 2° structures. In addition to determining biochemical substrate selectivity in vitro deaminase activity assays, the authors also analyzed mutations in mouse and human tumors to determine whether the innate substrate preference of these two enzymes is influenced by cellular factors.

The authors first investigated A3B vs A3A preference for deaminating cytidines within hairpin loops of varying loop size vs linear ssDNA, determining in conflict with a previous report from the same group (<https://www.science.org/doi/10.1126/science.aaw2872>) that A3B also prefers hairpins over linear ssDNA. The authors observed higher A3B activity on larger loops of 4 and 5nt, whereas A3A prefers smaller, 3nt loops (consistent with previous reports). By changing the positioning of cytidine within these larger loops, the authors were able to further conclude that A3B prefers 3' cytidine positioning. A3A was also found to deaminate larger loops with less efficiency but displays a marked preference for cytidine at position 4 within 5 nt loops in vitro. Results obtained using whole cell lysate was recapitulated using purified A3A and A3B. The authors also analyzed A3B- versus A3A-induced mutations in mouse tumors and observed similar respective substrate preferences as in their previous experiments, however A3B substrate preference appears to be dampened in this model, both in terms of preferring larger loops and requiring 3' C positioning within these loops. The authors concluded that this is likely due to the enhanced stability of small loop-forming hairpins in cells and cited cellular factors as a likely explanation for this observation. The authors also analyzed APOBEC mutations identified from whole genome sequencing of human tumors and further analyzed sequence context of mutations in tumors previously identified to be A3B vs A3A-preferred (using previously established preferences for R or Y respectively at the -2 position), and observed similar preferences for small hairpin loops with 3' C

positioning for both enzymes as those observed in mouse tumors.

The authors investigated how differing structures between A3A and the C-terminal domain of A3B influence their different substrate preferences by replacing loop 1 (located in close proximity to both enzymes' active sites) of A3B with the more truncated/restrictive loop 1 of A3A and observed a stronger preference for smaller hairpin loops as a result. Conversely replacement of A3A loop 1 with that of A3B significantly reduced activity on smaller hairpin loops but increased activity on larger loops resembling A3B's preferred substrate. The authors concluded that loop 1 plays a critical role in influencing the preference of these two enzymes for different substrates.

The authors also expanded on previous APOBEC literature by extending their analysis of mutations caused by A3B or A3A to the 5' end of the canonical APOBEC mutation signature to identify distinct signatures of each enzyme. Notably the authors used a deconvolution method, testing A3B and A3A activity on hairpin-forming oligonucleotides, systematically altering bases at the -2, -3, and -4 positions (in combination). Using this method they were able to identify a clear preference of A3B and A3A for loops where cytidine is preceded by a YYRT and YYYYT motif respectively (where Y is a pyrimidine and R is a purine), with a strong preference for C at the -4 position for both enzymes.

This manuscript provides a moderate advance in the field of APOBEC cancer biology by further defining A3B and A3A substrate specificity. The methodologies used were sound and most experiments were adequately rigorous, with reasoned conclusions being made, however statistical comparisons are generally lacking throughout the manuscript.

We thank the reviewer for his/her appreciation of the significance and quality of our work. We have now addressed all reviewer's comments. We would like to apologize for the lack of statistics in our first submission. We have now added additional statistical analysis throughout the manuscript.

Specific critiques:

1. Little to no rationale is given to explain why the previous report (<https://www.science.org/doi/10.1126/science.aaw2872>) indicated that A3B preferred linear sequences over hairpin.

The reviewer raised an important point. In our previous study, we showed that A3B has a similar preference for a hairpin DNA (with a 4-nt loop and a GT sequence before the TpC motif) over linear DNA (Buisson et al. Science, 2019 Figure 2D and Supp Figure 3). However, at that time, we were not aware of which sequences inside the loop of the hairpin would impact A3B activity. New results from our Oligo-seq assay now revealed that A3B dislikes GT sequence before the TpC motif of a 4-nt loop hairpin (**Figure 2F and 2I**), explaining why in our original study we did not observe that A3B targets stem-loop DNA over linear DNA. This further stresses the importance of developing an

unbiased method like Oligo-seq to study APOBEC activity. However, our original result is still valid since we confirmed in this manuscript that A3B has poor deaminase activity for a 4-nt loop hairpin with a GT sequence before the TpC site (**Supplementary Figure 3A**). We have now added additional information in the manuscript to compare this manuscript to our previous study:” *Note that these results explained why we did not previously report a preference toward hairpin DNA for A3B. It has now come to light that the 4-nt loop hairpin with a 5-GT sequence preceding the TpC motif, which was used in our previous study (16), proves to be a poor substrate for A3B compared to other sequences (Figures 2F, H-I, and Supplementary Figure 3A). This further stresses the importance of developing unbiased approaches such as Oligo-seq to study APOBECs activity.*”

2. Please include a gel (i.e. Coomassie, silver stain) that indicates purity of the purified A3A and A3B enzymes as contaminating proteins may influence enzyme substrate preference.

We have now added validation of the pull-down experiments by immunoprecipitation from human cell extract aimed to separate A3A and A3B from other soluble cell extract proteins. We particularly focused on RPA due to its ability to bind ssDNA and potentially impact A3B deaminase activity (**new Supplementary Figure 4A-B**). To properly distinguish that A3A and A3B used in the supplementary Figure 4 were not obtained by recombinant protein purification we have now changed the term “purified” to “pulled-down” in the Supplementary Figure 4 labels.

3. Please add a better description of what is being compared for the “relative mutation frequency” in the y-axis on figures 6 a,b,d, and e to the figure legend.

Sorry for the lack of explanation. The relative mutation frequency was calculated from the ratio of the number of mutations to the number of available sites, and normalized so that the background mutations rate (at all C:G positions) is equal to 1. We have now added this explanation in the figure legends.

4. A figure indicating how many A3A and A3B-induced mutations were observed in mouse and human tumors in hairpins versus non-hairpin forming sequences would be very helpful in evaluating the importance of hairpin mutagenesis in cancer. The data in Supplemental figure 8 c and d seem to indicate that hairpin mutations caused by A3B are very few.

As suggested by the reviewer, we have now measured the number of A3A and A3B-induced mutations in mouse and human tumors in hairpins versus non-hairpin forming sequences (**New Supplementary Table 1**). Although the levels of mutations are higher in the non-hairpin forming sequences, when normalized to the number of possible mutation sites found in the genome, the mutational frequencies in hairpins versus non-

hairpin forming sequences are similar (e.g., in human tumor, A3B has a mutation frequency of 1.45×10^{-4} in non-hairpin forming sequences versus 1.25×10^{-4} in hairpins).

5. If A3B is not making many mutations in mouse tumors, could the resulting lack of signal be one reason for the less prominent A3B hairpin signature in mice and human tumors, instead of other cellular factors (as suggested by the authors)?

This is a good point! A3A and A3B are known to have very different activity levels. In vitro, A3A has been shown to have a much higher deaminase activity than A3B (e.g., PMID: 23104058, PMID: 29234087). In vivo, it is now well established in the field that A3A caused higher levels of mutations in tumors than A3B (e.g., PMID: 35859169). Our discussion on the influence of cellular factors was to suggest that in addition to the difference in A3A and A3B enzymatic activity, the formation and stability of hairpins within cells might also impact the abilities of both A3A and A3B to induce mutations in hairpin DNA. Because A3B generates less mutations than A3A, these factors might have even a stronger impact on the total level of A3B-induced mutations detected in hairpins.

6. On lines 459-462, the authors claim they can use the mesoscale preferences of A3A and A3B to identify which enzymes are active in patient tumors. An analysis comparing the utilization of the mesoscale characteristics versus the previously used YTCA versus RTCA sequence specificity should be used to determine if the mesoscale preferences have added value in this differentiation.

This is another important point. We have now performed the analysis suggested by the reviewer by comparing directly patient tumors with enriched YTCA or RTCA mutations to the level of hairpin mutation character (**New Supplementary Figure 9B-C**). As expected, these analyses showed a poor correlation between hairpin mutation character and YTCA or RTCA mutation levels. Only in patients with very high levels of mutations in YTCA motifs, we were able to detect a correlation with the A3A hairpin mutation character. Together, these results further highlight the importance of not relying only on YTCA and RTCA sequence contexts to identify tumors with mutations driven by A3A or A3B.

7. Do A3B targeted hairpins also cause recurrent mutations in human or mouse tumors like A3A targeted hairpins do? Are any A3B targeted hairpin mutations potential drivers or are they primarily passenger mutations too?

We looked at recurrent mutations in hairpins caused by A3B. However, due to the low levels of mutations detected in hairpins and the limited number of high A3B+ tumor samples, we were not able to detect recurrent mutations in these hairpins. However, only one mutation from 258 mutated hairpins identified in A3B+ tumor samples was detected in a driver gene (TAF1 gene [D700N]). This result suggests that mutated

hairpins by A3B similarly to A3A are primarily passenger mutations. Future studies will be required to analyze a larger number of tumor samples and provide a more robust statistical demonstration of these findings.

8. The utility of the detailed analysis of A3A and A3B substrate specificity appears limited in regards to translational cancer biology. However, the authors indicate that this information may be very useful for designing better base editors at the end of the discussion. This idea appeared to not be well integrated into the rest of the manuscript, but could provide a larger impact for the work.

We believe that gaining a deeper understanding of A3A and A3B substrate specificity is critical for future targeted therapies of tumors that accumulate APOBEC mutational signatures. In fact, our recent research demonstrated how the understanding of A3A-induced mutations in hairpin DNA enabled the identification of a novel mechanism of resistance to EGFR inhibitors in lung cancer (Isozaki et al. Nature, 2023). In the future, we envision using our new study to assess whether a tumor accumulates mutations caused by A3A or A3B before treating patients with specific A3A or A3B inhibitors (when available) and further suppress tumor evolution.

We agree with the reviewer that our oligo-seq assay could be very useful for designing better base editors in the near future. While we are actively pursuing this research direction, we also hope that the reviewer agrees that this work extends beyond the scope of the current study, which primarily aims to determine the mutations induced by A3A and A3B in tumors.

REVIEWER COMMENTS

Reviewer #1 (Remarks to the Author):

Sanchez, et al have submitted a revised manuscript detailing the hairpin preference for A3B activity which is a means to distinguish it from A3A activity. The authors considered all reviewer comments and addressed many of them with additional experiments, statistical analyses, or editing of the text. To that end, this manuscript is polished and does not require further revision. My concern remains, as Reviewer 3 also wrote, that the larger point of this subtle distinction between A3A and A3B hairpin preference does not constitute a substantial advance for the field of cancer biology and is likely too nuanced for a general audience.

Reviewer #2 (Remarks to the Author):

The Authors have successfully replied to all reviewers' comments.

A few times it appears "cells' ability/potential". I believe the apostrophe should not be present

Reviewer #3 (Remarks to the Author):

I appreciate the authors responses to my prior comments. I still have the following suggestions:

- 1) In supplemental figure 3A, a direct comparison of A3B activity on a GTC hairpin substrate to a GTC linear substrate would be helpful to confirm that the activity on the GTC hairpin is still equal or less than to the corresponding linear substrate. The current manuscript seems to only show that GTC hairpin substrates are worse than other hairpin substrates. It does not make the comparison to linear substrates which was data that originally indicated that A3B did not prefer hairpins in the Buisson et al. Science 2019 manuscript.
- 2) Does the new data supplied in supplemental table 1 indicate that in tumors A3B is not significantly enriched for hairpin sites? The authors indicate that the frequency of mutation per site is similar between non-hairpin and hairpin forming sites.
- 3) The new correlation data provided in supplementary figure 9B-C does not really address whether the new hairpin specificity for A3B could be used to better identify patient tumors mutated by A3B or A3A.

Correlation analysis at best would provide support that non-hairpin mutations in RTCA mutations co-occur with A3B preferred hairpin mutations. It does not measure the number of tumors that have an indication of A3B mutagenesis, A3A mutagenesis, or both. Could the authors try to assign tumors based on YTCA/RTCA motifs and the combination of YTCA/RTCA with hairpin character to see if higher numbers of tumors could be definitively assigned? Maybe a clustering analysis using multiple APOBEC specific metrics could also work?

4) The suggestion to integrate the potential to use the hairpin specificity of A3B for base editors into the manuscript was meant to add text mentions of this throughout the manuscript (primarily in the introduction) to give a better sense of the overall potential utility of the data, not to conduct additional experiments. The utility of A3B hairpin sequence specificity to cancer diagnostics is still to be determined and even the A3A specificity that the authors reference as an example of this utility is more dependent on A3A's ability to edit RNA than hairpins. A3A's RNA editing capabilities provides a much greater degree of differentiation between it and other APOBECs compared to the differences in DNA hairpin editing sequence preference.

REVIEWER COMMENTS

Reviewer #1 (Remarks to the Author):

Sanchez, et al have submitted a revised manuscript detailing the hairpin preference for A3B activity which is a means to distinguish it from A3A activity. The authors considered all reviewer comments and addressed many of them with additional experiments, statistical analyses, or editing of the text. To that end, this manuscript is polished and does not require further revision. My concern remains, as Reviewer 3 also wrote, that the larger point of this subtle distinction between A3A and A3B hairpin preference does not constitute a substantial advance for the field of cancer biology and is likely too nuanced for a general audience.

We thank the reviewer for his/her appreciation of the significance and quality of our work. We believe that this study will be in the interest of a broad audience by going beyond the simple biochemistry characterization of A3A and A3B's substrate preference. Notably, we propose leveraging this substrate characterization to distinguish for the first time with high confidence the A3B mutational signature from the A3A mutational signature in patient tumors. To date, it was very challenging to identify tumors that accumulate mutation driven by A3B. This distinction is critical for future studies focusing on therapy resistance mediated by A3A or A3B. With our newly identified signatures, it is now possible to determine which enzyme, A3A or A3B, drives resistance to specific therapies. Furthermore, the characterization of A3A and A3B DNA binding activity on hairpin DNA may have relevance for virologists too, considering that the intrinsic role of these enzymes is to protect cells against viruses and DNA hairpins are frequently formed in DNA viral genomes. However, future studies are required to assess whether A3A and A3B binding to viral DNA hairpins affects their replication. Finally, our described Oligo-seq assay holds great potential for researchers involved in designing improved base editors or for characterizing other DNA-modifying enzymes.

Reviewer #2 (Remarks to the Author):

The Authors have successfully replied to all reviewers' comments.

A few times it appears "cells' ability/potential". I believe the apostrophe should not be present

We thank the reviewer for his/her appreciation of our work. We have now corrected these typos.

Reviewer #3 (Remarks to the Author):

I appreciate the authors responses to my prior comments. I still have the following suggestions:

1) In supplemental figure 3A, a direct comparison of A3B activity on a GTC hairpin substrate to a GTC linear substrate would be helpful to confirm that the activity on the GTC hairpin is still equal or less than to the corresponding linear substrate. The current manuscript seems to only show that GTC hairpin substrates are worse than other hairpin substrates. It does not make the comparison to linear substrates which was data that originally indicated that A3B did not prefer hairpins in the Buisson et al. Science 2019 manuscript.

We have now performed the experiment suggested by the reviewer. We compared a linear DNA substrate with GTTC sequence and a hairpin DNA substrate with GTTC sequence (both DNA

sequences were the ones used in the Buisson et al. Science 2019 manuscript) and compared to a hairpin sequence (CATC) found to be highly deaminated by A3B from our Oligo-seq assay. Consistent with our previous study, we showed that A3B has similar deaminase activity toward both linear DNA and the hairpin DNA with GTTC sequence, while A3B exhibited a much strong activity (6-7 times fold increased) for the 4-nt loop hairpin with a CATC sequence (**New Supplementary Figure 3B**). Together this new result confirms our previous findings and further highlights the importance of the sequence surrounding the TpC in the hairpin loop.

2) Does the new data supplied in supplemental table 1 indicate that in tumors A3B is not significantly enriched for hairpin sites? The authors indicate that the frequency of mutation per site is similar between non-hairpin and hairpin forming sites.

Sorry for the confusion. Most of the hairpins that can be formed in the mouse or human genome with a TpC site within the loop are not good substrates for A3A or A3B. Only hairpins with a TpC site in the 3' position of the loop and with a specific loop length are efficiently deaminated by A3A and A3B as shown in **Figures 6 A, B, D, E**. Our original table did not include the TpC position since this result was included in **Figure 6**. To avoid further misunderstanding, we have now extended our table to include the level and the frequency of mutations not only in hairpin DNA that have a TpC site in the loop of 3-6 nt but also 1) in hairpin DNA with a loop of 3-6-nt and a TpC position in 3', 2) in hairpin DNA with a loop of 3-nt and a TpC position in 3' (best hairpins for A3A), and 3) in hairpin DNA with a loop of 4-nt and a TpC position in 3' (best hairpins for A3B) (**New Supplementary Table 1**). These new data indicate that the frequency of mutations caused by A3A or A3B is lower in non-hairpin sequences than in their respective optimal hairpin-forming sites.

3) The new correlation data provided in supplementary figure 9B-C does not really address whether the new hairpin specificity for A3B could be used to better identify patient tumors mutated by A3B or A3A. Correlation analysis at best would provide support that non-hairpin mutations in RTCA mutations co-occur with A3B preferred hairpin mutations. It does not measure the number of tumors that have an indication of A3B mutagenesis, A3A mutagenesis, or both. Could the authors try to assign tumors based on YTCA/RTCA motifs and the combination of YTCA/RTCA with hairpin character to see if higher numbers of tumors could be definitively assigned? Maybe a clustering analysis using multiple APOBEC specific metrics could also work?

We apologize for misunderstanding reviewer's previous question. To determine whether the hairpin mutation character can be used to identify patient tumors mutated by A3A or A3B, we monitored the levels of YTCA/RTCA mutation character versus the levels of hairpin mutation character in each patient tumors. We then applied thresholds to classify patient tumors that accumulated high levels of YTCA mutations (100 patient tumors), RTCA mutations (70 patient tumors), A3A hairpin mutation character (343 patient tumors), or A3B hairpin mutation character (122 patient tumors) (**New Supplementary Figure 9D**). Individually, the two methods associated APOBEC-induced mutations with partially distinct subsets of patients' tumors and the hairpin mutation character method led to the identification of a higher number of patient tumors. Importantly, by combining both methods, 93 patient tumors were identified to be highly mutated by A3A and 30 patient tumors to be highly mutated by A3B. Together, these results highlight the importance of using both methodologies (as also indicated in **Figure 6F**) to determine with high confidence whether A3A or A3B drive mutations in a specific patient tumor. We thank the reviewer for suggesting this new analysis!

4) The suggestion to integrate the potential to use the hairpin specificity of A3B for base editors into the manuscript was meant to add text mentions of this throughout the manuscript (primarily in the introduction) to give a better sense of the overall potential utility of the data, not to conduct additional experiments. The utility of A3B hairpin sequence specificity to cancer diagnostics is still to be determined and even the A3A specificity that the authors reference as an example of this utility is more dependent on A3A's ability to edit RNA than hairpins. A3A's RNA editing capabilities provides a much greater degree of differentiation between it and other APOBECs compared to the differences in DNA hairpin editing sequence preference.

We apologize again for misunderstanding reviewer's previous question. As suggested by the reviewer, we have now added a section in the introduction about the use of APOBEC enzymes in base editor systems and the importance of understanding their substrate preference.